# *Clostridium difficile* Induced Inflammasome Activation and Coagulation Derangements

**DOI:** 10.3390/microorganisms10081624

**Published:** 2022-08-10

**Authors:** Marta Mattana, Riccardo Tomasello, Claudia Cammarata, Paola Di Carlo, Teresa Fasciana, Giulio Giordano, Alessandro Lucchesi, Sergio Siragusa, Mariasanta Napolitano

**Affiliations:** 1Department of Health Promotion, Mother and Child Care, Internal Medicine and Medical Specialties (ProMISE), University of Palermo, 90141 Palermo, Italy; 2Division of Internal Medicine, Hematology Service, Regional Hospital “A. Cardarelli”, 86100 Campobasso, Italy; 3Hematology Unit, IRCCS Istituto Scientifico Romagnolo per lo Studio dei Tumori (IRST) “Dino Amadori”, 47014 Meldola, Italy

**Keywords:** *Clostridium difficile*, induced inflammasome activation, coagulation derangements, enterocolitis, sepsis

## Abstract

*C. difficile* enterocolitis (CDAC) is the most common hospital infection, burdened by an increased incidence of coagulation-related complications such as deep vein thrombosis (DVT) and disseminated intravascular coagulation (DIC) as well as a significant sepsis-related mortality. In this review, we analyzed the available data concerning the correlation between coagulation complications related to *C. difficile* infection (CDI) and inflammasome activation, in particular the pyrin-dependent one. The little but solid available preclinical and clinical evidence shows that inflammasome activation increases the risk of venous thromboembolism (VTE). As proof of this, it has been observed that in vitro inhibition of the molecules (e.g., tissue factor) mainly involved in coagulation activation could block the process. In vivo studies show that it could be possible to reduce the incidence of complications associated with *C. difficile* infection (CDI) and mortality due to a state of hypercoagulability. A personalized therapeutic approach to reduce the inflammatory activity and prevent thromboembolic complications could be preliminarily defined to reduce mortality.

## 1. Introduction

*C. difficile* enterocolitis (CDAC) is the result of *C. difficile* infection (CDI), when the bacterium reaches a sufficient burden within the lumen of the colon. CDAC is the most common nosocomial infection whose manifestations can range from profuse diarrhea to surgical emergencies such as megacolon and volvulus [1]. One of the most dreaded complications of CDAC is sepsis, which in some cases can develop into septic shock with a secondary high mortality rate [2]. In this context, thromboembolic complications play an important but poorly explored role, thrombotic events have been associated with a three-fold increase in mortality secondary to CDI-related sepsis. The association between CDI and an increased risk of venous thromboembolism (VTE) is not yet fully understood and seems to find its pathogenesis in the activation of the inflammasome complex. Although the role of the inflammasome in the so-called ‘immune-thrombosis’ is now widely explored, most of the available evidence comes from in vitro studies. Several experiments in cellular and mouse models have shown that it is possible to block the hyperactivation of platelet aggregation and coagulation by acting on the inflammasome, but there are no viable therapeutic options yet. Several studies have been conducted on the ability of *C. difficile* to activate the inflammasome leading to a high incidence of VTE in the course of CDI, but studies providing insights into treatment and prophylaxis options in this context are not yet available. The primary aim of this narrative review (NR) was to analyze the scientific data, available to date, on the correlation between inflammasome and coagulation derangements within the course of CDI. The secondary aim of this review was to highlight currently available and future options of treatment and prophylaxis of venous thromboembolism caused by CDI-induced inflammasome activation, based on the available preclinical and clinical evidence.

### 1.1. Clostridium difficile

#### 1.1.1. Microbiology

Clostridium difficile (*C. difficile*) is a Gram-positive, spore-forming, anaerobic bacillus, which is widely distributed in the intestinal tract of humans and animals but also in the external environment. Transmission of this pathogen occurs by the fecal–oral route [3]. *C. difficile* produces two types of toxins, A and B (TcdA and TcdB), which are both enterotoxic and cytotoxic; some *C. difficile* strains are able to produce a third toxin, called *C. difficile* transferase (CDT). Toxins are transported to the cell cytoplasm, where they inactivate the Res-homologous (Rho) family of GTPases. The Rho protein takes part in actin polymerization, and therefore stabilizes the cell cytoskeleton [4]. TcdA and TcdB induce glycosylation of the Rho GTPases. In this form, Rho is inactive and the NFkB pathway is inhibited. This is a mechanism of immune escape. Rho inactivation is also associated with lower secretion of Toll-like receptor (TLR)-induced inflammatory cytokines and chemoattractants (Figure 1) [5]. The pathophysiological mechanisms by which *C. difficile* toxins are able to induce necrosis of the intestinal epithelium is not entirely understood. Given the excellent results, however, secured by treatment against *C. difficile* with fecal transplantation, it can be inferred that the intestinal microbiota has a crucial role in the cytotoxic effect of *C. difficile* [6].

#### 1.1.2. Clostridium difficile Infection

CDAC is the most common nosocomial infection mainly due to the administration of several antibiotic agents leading to a dysregulation of the colon microflora. This risk of CDAC is particularly increased by the exposure to broad spectrum antibiotics like penicillins and cephalosporins [7]. Other risk factors are older age and prolonged hospitalization [8]. The unbalance in gut microflora favors the maturation of *C. difficile*’s spores that are physiologically present in the colon lumen. The main symptom of CDAC is severe diarrhea (colitis) that causes electrolytic, plasma proteins (i.e., albumin), and fluid losses leading to hypovolemic shock. In particularly frail patients, the disruption of colon mucosa may allow the passage of bacteria from the intestinal lumen to blood vessels, thus causing sepsis and septic shock [9]. Other severe complications of CDAC include toxic megacolon, perforation, intestinal paralysis, kidney failure, and systemic inflammatory response syndrome (SIRS) [10]. CDAC should be ruled out in any case of diarrhea onset in a patient treated with antibiotics. This finding should prompt a test with enzyme immunoassay (EIA) of a stool sample to detect *C. difficile* toxins. It is also possible to detect *C. difficile* by testing its antigens with high sensibility and specificity, but this test does not detect toxins. Amplification of toxin genes by PCR is a highly sensitive and specific technique; however, it cannot tell whether the *C. difficile* strain is producing toxins or not [11]. *C. difficile* expresses surface layer proteins (SLPs or S-layer), composing a reticulum that contacts toll-like receptor 4 (TLR4), expressed on the host cell surface. S-layer binding to dendritic cells initiates downstream signaling of the nuclear transcription factor kappa β (NF-kB) and interferon regulatory factor 3, resulting in the production of inflammatory cytokines and the activation of immune cells. In vitro studies have also shown that signaling depends on the myeloid differentiation regulator gene 88 (MyD88), which is involved in neutrophil recruitment. In fact, TLR4-deficient and MyD88-deficient mice were more susceptible to infection and showed a worse clinic profile than controls. This displays that neutrophils are crucial in preventing bacterial dissemination through the damaged mucosa. In addition to the superficial proteins, the flagellum protein of *C. difficile* is also an activator of the immune response. The flagellins of b-proteobacteria and g-proteobacteria are in fact recognized by TLR5, which is expressed by enterocytes and resident dendritic cells. Activation of TLR5 decreases the secretion of immunoglobulin A and inhibits the generation of regulatory T-cells (Figure 2). Besides TLRs, sensors of intracellular innate immunity, including the nucleotide-containing intracellular domain 1 (NOD1) and the IL-1β/inflammasome domain, are also activated during CDI. In response to CDI, Nod1-deficient mice showed a decreased chemokine production, lower neutrophil recruitment, a higher bacterial translocation, and higher mortality than controls [12].

#### 1.1.3. Treatment and Prevention of *C. difficile* Enterocolitis

Treatment of CDAC is strictly related to the severity of the colitis. Non-severe forms are commonly treated with intravenous (i.v.) metronidazole. Severe CDAC, defined by hypoalbuminemia, leukocytosis, and acute kidney failure, on the contrary requires treatment with high i.v. dose vancomycin or extended oral fidaxomicin. The concomitant administration of other antibiotics during treatment for CDAC is discouraged because it is associated with prolonged diarrhea, and it may eventually lead to recurrent infections due to a dysregulation of the colon microflora [13].

Recurrence of CDAC is difficult to identify, it results in a high risk of serious complications or death and implies high costs for the healthcare system, in a single-center Romanian study conducted on all patients admitted for profuse diarrhea following antibiotic therapy, it was observed that patients with a higher incidence of CDAC recurrence were more likely to be older than 60 years and/or have a higher Charlson Comorbidity Index (CCI). Among the CCI comorbidities, the most frequently associated with CDI recurrence were the following: cardiovascular disease, gastro-intestinal diseases, dementia, and exposure to immunosuppressive treatments [14]. Prevention of CDI recurrence is not clearly defined but major progress has been made on the development of treatment strategies in this field. Bezlotoxumab is a monoclonal antibody directed against toxin B of *C. difficile*, it has been tested in combination with antibiotic therapy, showing a reduction of CDI recurrence in the placebo-controlled studies MODIFY I and II. Bezlotoxumab is currently included in the standard clinical care of CDI [15]. The potential role of probiotics in preventing the recurrences of CDI is yet unknown [16].

### 1.2. The Inflammasomes

#### 1.2.1. Structure of the Inflammasome

Inflammasomes are multiprotein cytosolic oligomers of the innate immune system mainly involved in the activation of inflammatory responses, against pathogens and other damage-associated signals [17]. Inflammasome requires a structural modification of the cytoskeleton, involving microtubules, of the neutrophils and macrophages in which the complex is formed [18]. The inflammasome typically consists of the following components: pattern recognition receptors (PRRs), the apoptosis-associated adaptor speck-like protein containing a caspase recruitment domain (ASC), and a zymogen that activates pro-inflammatory caspases (pro-CASP1). Inflammasome formation begins when the cytosolic sensor senses pathogen-associated molecular patterns (PAMPs), expressed by pathogenic microbes, damage-associated molecular patterns (DAMPs), expressed by host cells during cellular damage, stress signals, or cell death [19]. PRRs recruit ASC, which links the sensors to the effector pro-CASP1, this last cleaves into active caspases leading to the activation and assembly of the inflammasome, thus promoting proteolytic cleavage, maturation, and secretion of the pro-inflammatory cytokines interleukin 1β (IL-1β) and interleukin 18 (IL-18), as well as Gasdermin-D cleavage [20]. Gasdermin-D (GSDMD) is the substrate of pro-inflammatory caspases involved in inflammasome formation. Under resting conditions, GSDMD consists of an N-terminal subunit linked to a C-terminal subunit that acts as an autoinhibitor. The pro-inflammatory caspases cleave the bond between the two subunits, thus leaving the GSDMD-N free to bind to the plasma membrane and form a pore. Pore formation on the cell membranes causes osmotic swelling of the cell and eventually pyroptosis [21]. Pyroptosis and IL-1β secretion itself are mediated by Gasdermin D (GSDMD). This is shown by the fact that deletion of GSDMD in the mouse model of familial Mediterranean fever (FMF) results in the extinction of the inflammatory response and in the complete prevention of systemic complications and tissue damage effects mediated by IL-1β attraction of neutrophils and macrophages. GSDMD is crucial for the downstream production of IL-1β-dependent inflammatory chemokines such as IL-6, TNF, IL-18, IL-17A, RANTES, MCP-1, IFN-γ, IP-10, and G-CSF (Figure 3) [22]. Recently, a study conducted on irradiated mouse models of pulmonary infestation by *Aspergillus fumigatus* demonstrated that levels of GSDMD higher than normal were maintained even after the clinical remission of the aspergillosis [23]. There are several types of inflammasomes. The canonical inflammasome can be nucleated by the NLR (nucleotide-binding oligomerization domain and leucine-rich repeat-containing receptors) members NLRP1, NLRP3, and NLRC4 or by the ALR member AIM2. Additional NLR family members (e.g., NLRP2, NLRP6, NLRP7, and NLRP12) and the ALR family (e.g., NLRP1, NLRP3, and NLRC4, IFI16) have been demonstrated to have the ability to assemble the inflammasome, but the biological functions of these inflammasomes are not well-established and further studies are necessary to validate their real impact in the setting of the inflammatory response and disease [22].

#### 1.2.2. NLRP3

The most studied inflammasome is NLRP3, it is activated by a variety of different triggers, which include PAMPs, DAMPs, pore-forming toxins, crystals, and nucleic acid [24]. The NLRP3 inflammasome is essential for host immune defenses against bacterial, fungal, and viral infections; however, when dysregulated, it has also been linked to the pathogenesis of several inflammatory disorders including cryopyrin-associated periodic syndromes (CAPS), Alzheimer’s disease, diabetes, gout, autoinflammatory diseases and atherosclerosis [25]. Therefore, the activation of the NLRP3 inflammasome must be precisely regulated to provide an adequate immune protection to avoid any damage to host tissues. Indeed, several mechanisms, including post-translational modifications, ubiquitination, phosphorylation, and interacting partners, have been identified to regulate NLRP3 inflammasome activation [26]. NLRP3 and NLRP1 are involved in endothelial dysfunction through the regulation of immune–inflammatory processes in arterial endothelial cells. Endothelial activation not only creates a platform for the recruitment and interaction of various immune cells, but it also allows platelets activation. The vWF receptor GPIbα is essential for the interactions between platelets and endothelial cells as well as leukocytes, it also mediates thrombosis progression. A relationship between NLRP3 and platelets activation has been shown. A better understanding of inflammation-induced thrombosis will support the identification of new effective therapeutic interventions [27].

#### 1.2.3. Pyrin Inflammasome

Pyrin, encoded by the MEFV gene, is an intracellular pattern recognition receptor that assembles inflammasome complexes in response to pathogen infections. Mutations in the MEFV gene have been linked to autoinflammatory diseases such as familial Mediterranean fever (FMF) or pyrin-associated autoinflammation with neutrophilic dermatosis (PAAND). Interestingly, pyrin does not directly recognize molecular patterns (pathogen- or host-derived danger molecules), but rather responds to disturbances in cytoplasmic homeostasis caused by infections. In the case of pyrin, these perturbations, recently defined as homeostasis-altering molecular processes (HAMPs), lead to RhoA GTPase inactivation (Figure 4) [28].

## 2. Methodology

### 2.1. Narrative Review

The form of NR was chosen for this paper. The NR model was chosen instead of a systematic review because it was considered as the most useful for obtaining a broad perspective on this topic. Available data are relatively scarce and often not comparable; thus, it was not possible to integrate different studies, overcoming problems related to small sample sizes, resolving controversies arising from discordant studies, and increasing the generalizability of results, which are the main goals of a systematic review [29]. Given the overall diversity of the topics and study approaches included in the current analysis, a narrative review allows to link the relevant data presented in the papers in order to attempt to support a new theory [30,31].

### 2.2. MeSH Terminology

A non-systematic review of the literature available on electronic databases (PubMed, Scopus, Google Scholar) was performed. The following MeSH terms were used as search keywords: “*Clostridium difficile*”, “*Clostridium difficile* enterotoxin A”, “*Clostridium difficile* enterotoxin A receptor”, “*Clostridium difficile* enterotoxin B”, “*Clostridium difficile* enterotoxin B receptor”, “cdta protein, *Clostridium difficile*”, “inflammasomes”, “pyrin”, “NLRP3 protein, human”, “caspase 1 protein, human”, “caspase 4 protein, human”, “caspase 5 protein, human”, “inflammation mediators”, “cell, vascular endothelial”, “deep vein thrombosis”, “coagulation, intravascular disseminated”, “human microbiome”.

### 2.3. Analysis of Literature

A total of 130 papers published in a period ranging from 1992 to 2021 were analyzed through criteria of inclusion and exclusion. Of the 125 papers analyzed, 98 were included in the current review while 32 were excluded due to low pertinence to the original topic. The following inclusion criteria were applied to ensure the appropriateness of each paper to the chosen topic: “laboratory data on *C. difficile*-induced inflammasome activation, inflammasome-induced alterations in hemostasis mechanisms and *C. difficile*-induced alterations in hemostasis mechanisms”, “clinical evidence on *C. difficile*-induced alteration in hemocoagulation parameters and reports of CDI-associated thromboembolic complications”, “laboratory and clinical reports on the use of inflammasomes-inhibiting therapies and future perspectives on these therapies”. Exclusion criteria were applied as follows: “clinical reports on thromboembolism in the course of CDI but unrelated to *C. difficile*”, “CDI complications unrelated to inflammasome activation and specific therapies for CDAC unrelated to the inflammasome and thromboembolic complications”.

## 3. Preclinical Evidence

### 3.1. *C. difficile* Infection-Induced Inflammasome Activation

CDI is induced by a high bacterial burden that *C. difficile* is able to reach in the colon mucosa due to alterations in the quantitative and qualitative composition of the normal bacterial population, frequently induced by intense and/or prolonged treatment with antibacterial drugs [32]. Once *C. difficile* reaches enterocytes, its toxins (TcdA, TcdB, and CDT) are released into host cells. TcdA and TcdB act as glycosyltransferases able to block the function of different G proteins of the RHO family involved in regulation of cytoskeleton, progression of the cell cycle, and apoptosis [33]. On the other hand, CDT causes actin depolymerization by inducing alterations in the microtubule formation process. In fact, pyrin activation is upregulated by cytoskeleton structural changes [34]. Thus, the synchronous action of the three toxins on the cytoskeletal structure of the host cell causes the formation of the inflammasome complex. In antigen-presenting cells (APC) and leukocytes (neutrophils and macrophages), the inflammasome maintains a constant production of inflammatory cytokines and promotes cell apoptosis. Inflammation, sustained by the inflammasome, tends to evolve on a systemic scale in a time-dependent fashion [35]. Inflammatory cell recruitment and sepsis cause an overactivation of the coagulation cascade and platelet aggregation. These events contribute to an approximately three-fold increase in mortality during sepsis secondary to CDI. The impact of the coagulation process on the risk of death has been only partially explored in animal models in which pharmacological or genetic inactivation of the factors activating coagulation has been shown to reduce mortality [36].

#### 3.1.1. *C. difficile* Infection Mechanism Leads to Inflammasome Activation

*C. difficile* mechanisms of infection involve the binding of SLPs with cholesterol-rich microdomains, known as lipid rafts, in the host’s cell membrane. SLPs are divided into a stable high molecular weight protein and a variable low molecular weight protein, resulting from the cleavage of the product of the gene slpA. Following adhesion, TcdA and TcdB enter the host through protrusion induced by CDT. *C. difficile* adhesion to infected cells triggers the release of inflammatory cytokines like IL-1β and IL-23. Chen et al. demonstrated that SLPs induce the release of IL-1β and the maturation of caspase-1, thus activating the inflammasome [37]. *C. difficile* SLPs activate the inflammatory response through binding to the toll-like receptor 4 (TLR4) of enterocytes. This causes the activation of the transcription factor nuclear factor kB (NF-kB), which induces the transcription and translation of genes coding for inflammatory cytokines whose task is to recruit leukocytes to the site of infection. Prominent among these cytokines are IL-1β and IL-18, the production of which occurs through the formation of the pyrin inflammasome by the binding of pyrin to the adaptor ASC [12]. Inflammasome activation is an important mediator of the innate immune response that requires a priming and an activation phase following which a cleavage occurs, thus leading to a particular type of cell apoptosis, called pyroptosis, with release of inflammatory cytokines by dead cells. In general, the main factors that trigger inflammasome activation are the recognition of PAMPs and DAMPs or the action of bacterial toxins. In the context of CDI, the pyrin inflammasome can be activated by sensing the inactive state of Rho GTPases, which are in turn inactivated by toxins A and B. The action of the inflammasome results in the release of pro-IL-1β and IL-8 from the pores of the cytoplasmic membrane [38].

#### 3.1.2. Role of *C. difficile* Toxins in Inflammasome Activation

TcdA and TcdB, as well as several other bacterial toxins, indirectly activate the pyrin inflammasome by inhibiting the GTPase activity of the Rho protein, without which there are no longer free phosphate groups available to keep pyrin phosphorylated. Dephosphorylated, pyrin can bind the ASC adaptor and activate caspase-1, allowing the formation of the inflammasome [39]. Furthermore, TcdA and TcdB contribute to the release of IL-23 from dendritic cells by yet unknown mechanisms. The toxins also synergize with MyD88 in the production of IL-23. MyD88 mediates the PAMPs’ and DAMPs’ danger signals. The activation of these signaling pathways increases production of IL-1β while also promoting the expression of IL-1 receptor (IL-1R) [40]. Finally, TcdA and TcdB induce transcription of the MEFV gene encoding pyrin [41]. TcdA, in particular, is able to glycosylate Rho. However, this seems to be related to the uncontrolled activation of the pyrin inflammasome only in neutrophils, as monocytes lack the ASC adaptor, so inhibition of Rho does not have the same effect. Taking into account the inflammatory potential of TcdA’s glycosylating activity on Rho, a new therapeutic frontier might be to inhibit Rho [42].

#### 3.1.3. Alternative Apoptosis Pathways in Response to *C. difficile* Infection

Pyroptosis of enterocytes is the primary mechanism of defense against CDI, but it also causes diarrhea and loss of integrity of tissues, thus, allowing other bacteria penetration into blood flow and sepsis. This excessive inflammatory response is largely sustained by the constant production of IL-1β. Pharmacological inactivation of IL-1β by Anakinra has been shown to reduce mucosal damage ex vivo [43]. Although the role of the pyrin inflammasome in the genesis of the inflammatory response to CDI seems predominant, it has been shown that the pathway of the ATP-P2X7 (adenosine triphosphate—purinergic receptor 2X7) complex, known as an activator of NLRP3, is essential for the inflammatory response itself. Inhibition of P2X7, in fact, completely turns off inflammation. The ATP-P2X7 complex is formed when intracellular ATP is released from enterocytes, neutrophils, and monocytes that have undergone various types of apoptosis. This substrate binds P2X7, forming a complex capable of immediately activating caspase-1 and inducing pyroptosis (Figure 5). P2X7 is also involved in cellular antibacterial defense mechanisms such as the formation of phagolysosomes in monocytes and the release of oxygen free radicals (ROS) [44]. CDI is the leading cause of antibiotic-associated diarrhea, currently representing a significant health burden. Although the role and contribution of *C. difficile* toxins to the pathogenesis of the disease is increasingly understood, other aspects of *C. difficile*–host interactions, particularly the effects of the bacterium on host immunity, are currently less well studied. Using an ex vivo infection model, it has been reported that the human gastrointestinal mucosa elicits a rapid and significant cytokine response to *C. difficile*: a marked increase in IFN-c with a modest increase in IL-22 and IL-17A. The significant increase in IL-8 suggested a potential influx of neutrophils, while the presence of IL-12, IL-23, IL-1β, and IL-6 suggested a cytokine environment able to modulate subsequent T-cell immunity [45]. *C. difficile* toxins A and B are certainly the major causes of pyrin inflammasome activation in neutrophils and monocytes; however, intestinal epithelial cells (IECs) are the primary target cells of *C. difficile* exotoxins. It is not yet known whether in vivo inflammasome signaling by pyrin affects the cytotoxicity of IECs induced by *C. difficile* or modulates the course of CDI pathophysiology. A study performed on primary organoid systems of IECs showed that *C. difficile*-induced cytotoxicity is completely mediated by its exotoxins and that pyrin inflammasome is not a necessary condition for IEC death. The death of IECs induced by TcdA and TcdB relies on caspase-3/7-mediated apoptosis via the intrinsic apoptotic pathway; however, blockade of caspase-8 death receptor-induced apoptosis and necroptotic signaling is not protective against cell death, as is the transgenic expression of the antiapoptotic protein Bcl-2 in intestinal organoids, this phenomenon suggests the existence of another mechanism of apoptosis [46]. It is currently unknown whether the cytotoxicity of IECs in the context of CDI is harmful or beneficial to the host [47]. In the knock-out mouse model for the caspase 3 and 7 genes, in IECs there is an increased susceptibility to CDI and a worse CDAC. Furthermore, the lack of pyrin expression in IECs protects these cells from pyroptosis upon exposure to *C. difficile* toxin and instead promotes apoptosis of IECs as the dominant mechanism of death during CDI in vivo. This evidence suggests that *C. difficile*-induced IEC apoptosis is an early host defense mechanism that contributes to bacterial restriction [46].

### 3.2. Inflammasomes and Thrombosis

#### 3.2.1. Inflammasome-Induced Coagulation Derangements

Inflammasome activation has been associated with thromboembolic events. This phenomenon finds its explanation in the close relationship between inflammation and thrombosis, two mechanisms that can interact by forming a vicious cycle. Specifically, inflammation can alter the balance between pro-thrombotic and antithrombotic factors, thus favoring thrombus formation, whereas thrombus can amplify the inflammatory response [46]. Moreover, the involvement of inflammasomes in the pathophysiology of several inflammatory disorders such as systemic lupus erythematosus (SLE), rheumatoid arthritis (RA), and inflammatory bowel diseases (IBDs) with prominent pro-thrombotic phenotypic features confirms the strong interaction between inflammation and coagulation. Activation of inflammasomes has been shown to act as an independent risk factor for thrombosis in patients with SLE. Similarly, recent work has shown an increased expression of NOD2, NLRP3, and NLRC5 in cytoplasmic antineutrophilic antibody (ANCA) associated with vasculitis (AAV), compared to normal controls. The pathogenetic role of NLRP3 in IBDs has also been explored. Patients with IBD have an up to three times higher risk of thromboembolic complications associated with an increased morbidity and mortality [48]. The inflammasome promotes thrombus formation by recruiting immune system cells, endothelial cells, and platelets [49]. CDI is characterized by a degree of absolute neutrophil count in peripheral blood, this finding is highly related to sepsis and is considered as an indicator of poor prognosis. IL-23 release can be enhanced by neutrophils through the activation of NF-kB [19]. The importance of inflammasome and its potential role in thrombosis has been highlighted in a recent study describing how NLRP3 activation increases the risk of venous thromboembolism (VTE) in response to hypoxia. Using a murine model of venous thrombosis, the authors showed that systemic hypoxia is associated with the activation of the NLRP3 inflammatory complex and the production of IL-1β, which accelerates the development of venous thrombosis. Histological examination of thrombus showed increased recruitment of neutrophils and macrophages into the vein in mice subjected to hypoxia compared with controls [50]. The exact molecular mechanism by which the inflammasome exerts its influence in the pathophysiology of thrombosis still needs to be explored. Signaling pathways through which NLRP3 contributes to endothelial and platelet activation are still ambiguous. Therefore, a clear definition of the pathways targeted by NLRP3 inflammasome under hypoxia could provide the clue toward the integrated involvement of hypoxia-NLRP3 inflammasome and vascular dysfunction/hypercoagulation [51].

#### 3.2.2. Role of Endothelial Dysfunction

The vascular endothelium plays a crucial role in the regulation of homeostasis. NLRP3 and NLRP1 are responsible for endothelial dysfunction through the regulation of immune–inflammatory processes in arterial endothelial cells. A relationship between the role of NLRP3 and platelet activation has been shown. A better understanding of inflammation-induced thrombosis could support the identification of new effective therapeutic interventions. Some powerful and specific NLRP3 inhibitors such as MCC950 might be adopted in the management of this disease. Thus, the investigation of drugs that target the inflammasome may open up a completely new line of thrombosis treatments [52]. Furthermore, the inflammatory microenvironment that leads to endothelial dysfunction has recently been correlated to the activation of gasdermin and GSDMD formation in neutrophils in the setting of sickle cell disease [53]. This process may be seen as an independent risk factor for microembolism in the setting of CDI, too.

#### 3.2.3. Inflammasome-Induced Disseminated Intravascular Coagulation

Coagulation acts as a trap for bacteria by isolating them in the microcirculation where antimicrobial responses are triggered. This mechanism is virtuous in localized infections but becomes a vicious cycle in sepsis, as it escalates to a systemic involvement and leads to disseminated intravascular coagulation (DIC). Impaired anticoagulant proteins and endothelial dysfunction also contribute. DIC triples the mortality risk of patients in septic shock. Clinically, it has indeed been observed that heparin prophylaxis in patients in septic shock prevents disseminated intravascular coagulation (DIC) [54]. The molecular mechanisms of how systemic coagulation is triggered by the inflammasome during sepsis leads to a new understanding of inflammasome function and sets a new stage on immune coagulation. Indeed, during sepsis, the host immune response triggers coagulation activation by inducing tissue factor expression on monocytes, platelets, and endothelial cells [24].

### 3.3. Role of Leukocytes and Platelets in Inflammasome-Induced Thrombosis

#### 3.3.1. Neutrophil Recruitment Promotes Thrombosis during *C. difficile* Infection

Neutrophil recruitment in the intestinal tract following NF-κB signaling via the TLR4 and Nod1 signaling pathways is essential to protect the host from *C. difficile* [55]. Neutrophil activation leads to an increased production of active caspase-1 and a simultaneous formation of neutrophil extracellular traps (NETs). NETs are essential for caspase-1 activation on platelets; indeed, it was seen that platelets constituted more than half of the cells containing active caspase-1. Inhibition of caspase-1 significantly reduced DVT in studies on mice. Taken together, the presence of NETs/caspase-1 complexes and activation of caspase-1 in thrombi implies that these complexes are likely involved in DVT development [56]. NETs also appear to play a role in the progression to septic shock [57]. Neutrophils and monocytes are also reservoirs of tissue factor (TF) that are massively released following their death by pyroptosis within the course of the inflammatory response. The activation of FVII resulting from this event causes microthrombosis to occur at the site of infection, with secondary tissue hypoxia. Hypoxia is the major activator of caspase-1 and the NLRP3 inflammasome. This synergy between the pyrin inflammasome and the NLRP3 inflammasome has been shown to greatly favor DVT during infection and, therefore, suggests that inhibition of the inflammasome could prevent both septic shock and sepsis-associated DVT (Figure 6) [58].

#### 3.3.2. Role of Platelets

Platelets play an important role in the systemic inflammatory response to infections [59]. Indeed, one of the main causes of sepsis progression to septic shock is endothelial dysfunction, which is often caused by the release of IL-1β-containing platelet microparticles (PMPs) [60]. The production of substantial amounts of IL-1β within platelets is even caused by low concentrations of lipopolysaccharide (LPS) [61]. This effect appears to be much more powerful than the action of other platelet agonists such as thrombin or collagen [62]. NLRP3 represents the molecular link between inflammation and thrombosis; it is, in fact, expressed not only on inflammatory cells but also on platelets. Platelet NLRP3 is involved in the regulation of endothelial permeability in dengue infection. In vivo, NLRP3 has been shown to promote thrombus formation by inducing platelet aggregation. A decreased platelet aggregation and impaired clot retraction was observed in NLRP3-deficient platelets [63].

## 4. The “Other Side” of the Inflammation–Coagulation Link: Inflammasome and Fibrinolysis

A direct correlation between chronic inflammasome activation and an increased thromboembolic risk has been shown by several studies in murine and cellular models. Current findings highlight how a prolonged inflammatory state can promote platelet aggregation and thrombin generation through an imbalance between the coagulation cascade and physiological anticoagulant mechanisms. However, although less well-known, there is a correlation between chronic inflammation, sustained by the inflammasome, and an overactivation of fibrinolysis.

A potential link between inflammasome and fibrinolysis is the function of annexins. Annexins are proteins that activate tissue plasminogen activator (TPA), which in turn activates the fibrinolysis pathway. Overexpression of annexin A2 is in fact found in severe hemorrhagic diseases where fibrinolysis is the main feature of the coagulation abnormality, like in acute promyelocytic leukemia [64]. Following several triggering factors, inflammatory cells (i.e., macrophages and dendritic cells) are activated to induce lysosomal membrane damage, associated with recruitment of annexin A2. Annexin A2 deficiency correlates with an increased lysosomal damage, associated with the release of cytokines able to activate NLRP3 inflammasome, thus leading to the secretion of IL-1. It can be assumed that annexin A2 deficiency may increase thrombotic risk [65]. Studies on multiple sclerosis (MS) showed that missense and nonsense mutations in genes coding for fibrinolysis, complement (PLAU, MASP1, C2), angiogenesis and inflammasome proteins (NLRP1-2) may cause a derangement of the pathways associated with chronic inflammation. Several proteins of the fibrinolysis and complement cascade have been shown to regulate inflammation [66].

## 5. Clinical Evidence

### 5.1. Thrombin Generation Assay

The thrombotic risk of patients with CDAC was anecdotally studied. An observational study performed by Mihăilă RG et al. showed that patients with CDI had a specific pattern of thrombin generation (TG); the authors found that, in particular, two parameters of TG were higher than in the control group: mean velocity index and peak thrombin. These findings suggest an increased thrombotic risk during CDI even in the absence of septic shock. Thrombin generation assay was performed on a population of 84 individuals composed of patients with CDI (*n* = 33) and healthy volunteers (*n* = 51). Indeed, *C. difficile* was independently associated with venous thromboembolism [67]. TG assay is a novel diagnostic test capable of globally evaluating hemostasis. In particular, higher peak thrombin values have been associated with an increased risk of VTE [68]. The pathophysiology mechanisms explanation of this abnormality lies not only in the inflammatory process that triggers pro-thrombotic events but also in a predominant role played by the toxins of *C. difficile*, able to increase thrombin generation through the Ras-related C3 botulinum toxin substrate 1 (Rac-1) signaling mechanism. The evaluation of the individual thrombotic risk of patients with CDAC by thrombin generation assays to guide the tapering of thromboprophylaxis strategies may be a useful tool to prevent thromboembolic complications in selected cases. Thrombin generation may therefore contribute to a personalized prophylactic treatment [67].

### 5.2. Incidence of VTE in Hospitalized Patients with CDI

The correlation between CDI and venous thromboembolism has not yet been evaluated in studies on large populations; however, it has been reported in several case series. CDI has been observed to be an independent risk factor for VTE. Available real-life data on the administration of anticoagulant therapy or prophylaxis with low molecular weight heparin have shown an effective advantage in the survival of patients with CDI. New therapeutic agents acting on various checkpoints of the coagulation cascade and on the inflammatory process could interfere with the development of immune-thrombosis and are currently under study.

One of the first studies on the correlation between CDI and VTE was conducted in 1992 on sera samples of 10 patients diagnosed with CDI between 1982 and 1984. The study found that the permeabilization of the enteric mucosa and dysbiosis caused by CDI determines a reduced intestinal absorption of several molecules like albumin, protein S, Antithrombine III (ATIII), and vitamin K (VK). Vitamin K deficiency is associated with a reduction of clotting factor zymogens activation but also with a reduced activity of VK-dependent anticoagulant proteins like protein C and protein S. Thus, CDI can be associated with either bleeding or thrombotic complications [69]. More recently, in 2013, the American College of Chest Physicians along with US government agencies proposed a stratification of venous thromboembolic risk in hospitalized patients in order to study preventive strategies. In this study, performed by Merrill S et al., *C. difficile* test positivity was shown to be associated with an increased risk of DVT compared to *C. difficile*-negative patients: OR:3.23 (95% CI) vs. OR:1.95 (95% CI). This means that the risk of developing DVT, regardless of other causes, is doubled in patients with CDI when compared to *C. difficile*-negative patients [70]. Furthermore, an association between CDI and poor prognosis has been observed. Khushali et al., on the 2016 National Inpatient Sample (NIS), identified patients with VTE diagnosed with CDI: 3080 patients out of 382,585 were reported. A higher mortality during hospitalization (6% vs. 3%) and a longer hospitalization (13.7 days vs. 5.8 days) were found in DVT patients with CDI, compared to other patients without CDI. This study was the first to hypothesize an independent association between CDI and DVT [71]. Finally, a case report by Kumar et al. from 2021 showed an association between VTE or arterial thrombosis (cerebral stroke, arterial limb thrombosis) and CDI. The authors reported the case of an 82-year-old woman admitted to the emergency area for septic shock secondary to CDI. During the hospitalization, bilateral lower extremity venous thrombosis, pulmonary embolism, multifocal thromboembolic brain infarctions, and acute arterial thromboembolic occlusion of the right upper and lower extremities were detected [72].

## 6. New Frontiers of Treatment for CDI-Induced Immune-Thrombosis

Treatment for *C. difficile* enterocolitis includes antibiotic and supportive therapy. Although activation of the inflammasome is responsible for immune-thrombosis in these patients, modulating the inflammasome pathway has not yet been shown to effectively prevent the activation of coagulation in vivo. Furthermore, the administration of agents directly targeting the coagulation system (like TF inhibition) has never been tried in vivo. Consequently, there is no specific indication for prophylaxis of venous thromboembolic events in CDI in the absence of other known risk factors for venous thromboembolism.

### 6.1. Prophylaxis of CDI Recurrence and Its Thrombotic Complications

Although exposure to antibiotic treatment is a major predisposing factor for CDI, metronidazole and vancomycin are needed to treat CDI. These therapies continue to suppress the normal gut flora, thus CDI may recur in approximately 25% of cases. Since much of the effects induced by CDI result from the acute host inflammatory response stimulated by *C. difficile* toxins, it has been proposed that a combination of antibiotic and anti-inflammatory agents might be an effective treatment strategy. In agreement with this theory, a number of studies have shown that anti-inflammatory agents can reduce CDI severity in animal models of infection. Nevertheless, certain types of innate immune signaling may actually prevent *C. difficile*-induced immunopathology. Jarchum et al. recently showed that the administration of the TLR5-agonist Salmonella-derived flagellin was protective against *C. difficile*-induced colitis [73].

TLR5 signaling induced by intestinal microbiota stimulates productive innate immune responses that make mice resistant to CDI. This hypothesis is based on the observations that antibiotic pretreatment sensitizes mice to CDI, this pretreatment reduces TLR signaling in mice; furthermore, mice defective in innate immune signaling (MyD88-deficient) are more susceptible to CDI [74].

The most dreaded complication of immune-coagulation is DIC. The development of thrombosis involving several districts and the over consumption of coagulation factors in DIC is related to the activation of the coagulation cascade triggered by a massive release of TF from macrophages during pyroptosis. This mechanism has been proven by demonstrating that pharmacological or genetic inactivation of TF in vitro causes a complete blockade of the coagulation activation [75]. A retrospective study by Bhandari et al. on 27,545,773 adult patients aged 18 to 80 years who were hospitalized in 2011 found that 262,934 (1%) were positive for *C. difficile*. Among these subjects, 3% were diagnosed with DVT and 2% were diagnosed with PE. The authors suggested that, given the statistically significant increase in the risk of VTE, antithrombotic prophylaxis with low molecular weight heparin (LMWH) may be taken into account in patients with CDI [76].

Thrombomodulin may also be able to control the evolution of DIC. In a case report from Minami K et al., a 51-year-old man affected by DIC secondary to CDI was successfully treated with PMX-HP to improve septic shock and rhTM to control DIC. Many studies have shown that rhTM exerts an anti-inflammatory action through the neutralization of HMGB-1, a pro-inflammatory mediator produced by necrotic cells or macrophages/activated dendritic cells involved in shock and multiorgan failure. In conclusion, in patients with fulminant CDAC resulting in septic shock and DIC, the use of both PMX-HP and rhTM may positively affect survival [77].

TcdB is the main virulence factor of *C. difficile*. Luo J et al. performed a CRISPR/Cas9 screening and identified tissue factor pathway inhibitor (TFPI) as the receptor of TcdB4. TFPI is expressed by intestinal glands, so by administering recombinant TFPI, the colonic epithelium may be protected from *C. difficile* infection and its complications [78].

### 6.2. Inflammasome Inhibiting Drugs

The current understanding of the function of the inflammasome is limited to the inflammatory response. Therefore, by identifying DIC as the critical final event of inflammasome activation and pyroptosis, a new avenue in the study of these mechanisms is opened.

Most studies have focused on the direct effects that inflammasomes exert on tissue factor release and activation. Treatment with anticoagulants after the onset of sepsis did not result in improved clinical outcomes. However, the administration or combination of inflammasome inhibitors could be a favorable approach to prevent or early-treat sepsis-induced coagulation activation [79]. The administration of inflammasome inhibitors in diseases where inflammation plays an important role was evaluated in a study by Sánchez-Fernández A et al. The authors studied OLT1177 (Dapansutrile), a novel NLRP3 inflammasome inhibitory therapeutic agent, in a mouse model to prevent functional deficits in the course of encephalomyelitis. The study results suggest that the selective inhibition of inflammasome could reduce tissue damage mediated by inflammation [80]. Another molecule that has been proven effective in the inhibition of NLRP3 is oridonin (Ori). Ori is the active molecule of the Chinese herb *Rabdosia rubescens*, with anti-inflammatory properties. The compound has been used in a mouse model of peritonitis, gout, and type 2 diabetes mellitus with encouraging results [81].

Drugs that regulate NLRP3 inflammasome activation have already been adopted in the treatment of acute respiratory distress syndrome (ARDS) in patients with SARS-CoV-2 interstitial pneumonia. Our group has suggested that SARS-CoV-2 infection leads to hyperactivation of the NLRP3 inflammasome and that inhibition of inflammatory cytokine production by hydroxychloroquine may reduce the risk of developing ARDS and associated complications such as pulmonary fibrosis [82].

Direct inhibition of pro-inflammatory caspases via synthetic peptides is burdened by high toxicity and poor bioavailability. Among the new agents developed to overcome these problems are pralnacasan and belnacasan which act as caspase 4 and 5 inhibitors respectively. Further downstream in the inflammasome mechanism, on the other hand, the alkylating agent necrosulfonamide (NSA) acts to inhibit GSDMD pore formation. The same mechanism is shared by the agents disulfiram and Bay11-7082 [83].

MCC950, on the other hand, acts as an NLRP3 inflammasome inhibitor by blocking oligomerization and, thus, the functionality of the ASC adaptor [84]. MCC950 has already been shown to reduce endothelial dysfunction in sepsis models in vitro and in vivo [85]. Other noteworthy NLRP3 inhibitors are CY-09, OLT1177, 3,4,methylenedioxy-β-nitrostyrene (MNS), tranilast, and oridonin [86]. Among these, oridonin has been shown to prevent endothelial dysfunction by inhibiting MAPKs and the NF-kB signal transduction cascade [87].

In a mouse model of focal cerebral ischemia, intraventricular injection of anti-NLRP1 antibody was shown to inhibit the inflammatory response resulting from ischemia. This effect was observed by the substantial reduction in the concentration of mature IL-1β, resulting in an overall downregulation of the inflammasome function [88]. This finding was recently explored by a clinical trial where the administration of an IL-1β receptor antagonist brought encouraging results in the setting of stroke [89].

### 6.3. Clinical Trials

#### 6.3.1. Microbiota-Based Drugs

An attempt to prevent *Clostridium difficile* recurrence is represented by therapies based on the oral administration of bacterial spore formulations. Two phase 2 studies were conducted on this topic: two formulations called VP20621 and RBX2660 (formulation of microbiota-derived bacteria) were respectively administered, with poor results.

VP20621 consists of spores of the M3 strain of NTCD. In this clinical trial, healthy adult subjects aged 18–45 years or older than 60 years received single or multiple doses of an oral suspension of VP20621 (104, 106 or 108 spores) or placebo. The group of patients older than 60 years received oral vancomycin for 5 days, followed by VP20621 or placebo. VP20621 was well-tolerated and able to colonize the gastrointestinal tract even in subjects pretreated with vancomycin. Further studies of VP20621 to prevent CDI in patients are warranted [90].

The double-blind, randomized, placebo-controlled trial of the first microbiota-based drug included two doses of RBX2660 spaced apart vs. placebo. Results showed that two doses of RBX2660 were not superior to the placebo, but a single dose was significantly more effective than the placebo. Future clinical evaluations are required to confirm the long-term benefit and safety of RBX2660 [91].

#### 6.3.2. Eculizumab

Galic et al. reported a case of CDI treated with Eculizumab, a humanized anti-C5 monoclonal antibody. Eculizumab was administered on day 10 after diagnosis of CDI, with a marked improvement in the patient’s clinical condition, a restoration of the platelet count and other laboratory parameters. The risks and benefits of Eculizumab administration must be well-weighed. On the one hand, it may prevent tissue damage in the MOF; on the other hand, it carries an increased risk of meningococcal infection and encapsulated bacteria if administered when the infection is not yet well-controlled [92] (Table 1).

## 7. Conclusions

The complications induced by CDI on coagulation activation do not seem to be of particular clinical interest because they are often not recognized as being directly caused by the infection. Nevertheless, infection-related thromboembolic complications increase mortality in sepsis, especially in frail individuals with several comorbidities. Several preclinical studies have shown that interfering with inflammasome can control not only the infection but also the uncontrolled activation of hemostasis (immuno-coagulation) on several checkpoints. The future perspectives in this sense are to translate the biological and laboratory data to a clinical approach in order to improve the management of complications related to *Clostridium difficile* infection.

## Figures and Tables

**Figure 1 microorganisms-10-01624-f001:**
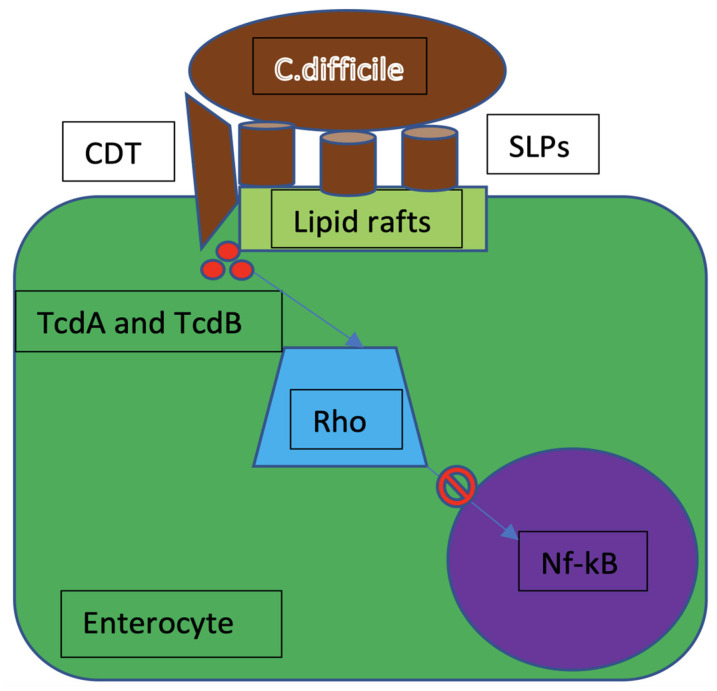
*C. difficile* mechanism of infection. CDT = *C. difficile* transferase; SLPs = surface layer proteins; TcdA = *C. difficile* toxin A; TcdB = *C. difficile* toxin B; Rho = rhesus homologous; Nf-kB = nuclear factor-kB.

**Figure 2 microorganisms-10-01624-f002:**
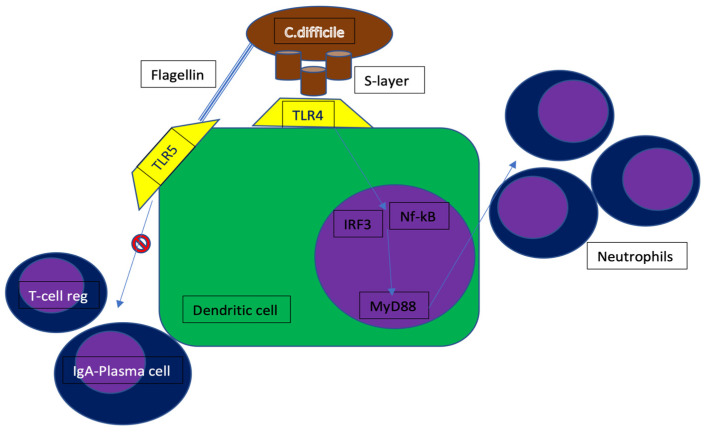
*C. difficile* mechanism of immune-escape. TLR4 = toll-like receptor 4; TLR5 = toll-like receptor 5; IRF3 = interferon regulatory factor 3; MyD88 = myeloid differentiation factor 88; Nf-kB = nuclear factor-kB.

**Figure 3 microorganisms-10-01624-f003:**
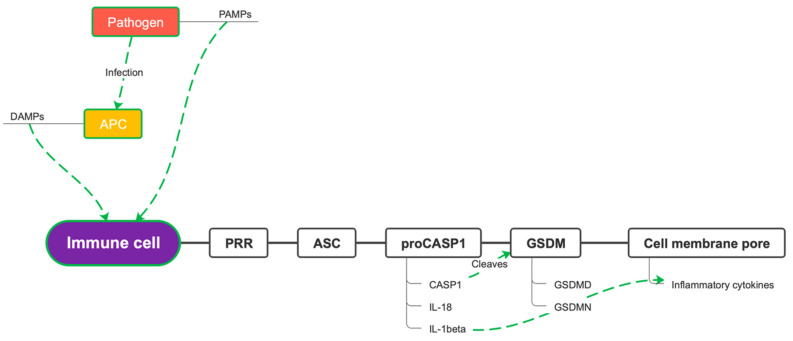
Gasdermin pore formation. PAMPs = pathogen-associated molecular patterns; DAMPs = damage-associated molecular patterns; APC = antigen-presenting cell; PRR = pattern recognition receptor; ASC = apoptosis-associated adaptor speck-like protein containing a caspase recruitment domain; pro-CASP1 = pro-caspase-1; CASP1 = caspase-1; IL-1 beta = interleukin-1 beta; IL-18 = interleukin-18; GSDM = gasdermin; GSDMD = gasdermin-D.

**Figure 4 microorganisms-10-01624-f004:**
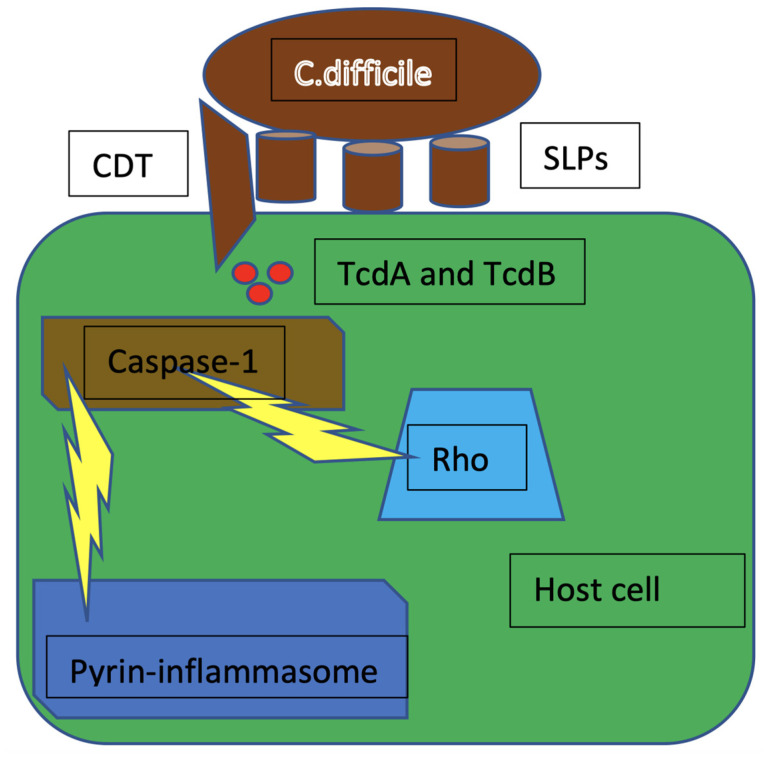
The inactivation of Rho by TcdA and TcdB causes cytoskeleton rearrangements that trigger pro-caspase-1 cleavage and activation, leading to the formation of pyrin inflammasome. CDT = *C. difficile* transferase; SLPs = surface layer proteins; TcdA = *C. difficile* toxin A; TcdB = *C. difficile* toxin B; Rho = rhesus homologous.

**Figure 5 microorganisms-10-01624-f005:**
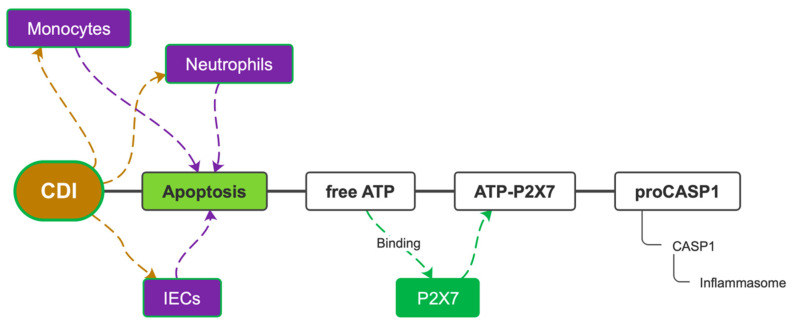
The ATP-P2X7 pathway. CDI = *C. difficile* infection; IECs = intestinal epithelial cells; ATP = adenosine triphosphate; P2X7 = purinergic receptor 2X7; pro-CASP1 = pro-caspase-1; CASP1 = caspase-1.

**Figure 6 microorganisms-10-01624-f006:**
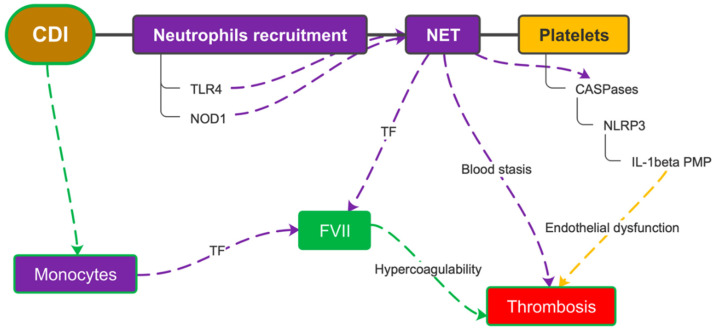
Leukocyte recruitment and pyroptosis due to CDI causes the release of tissue factor from monocytes and neutrophils. This acts in synergy with platelet activation and, together with the endothelial dysfunction caused by the inflammatory microenvironment, favors thrombus formation. CDI = *C. difficile* infection; TLR4 = toll-like receptor 4; NET = neutrophil extracellular trap; TF = tissue factor; NLRP3 = nucleotide-binding oligomerization domain and leucine-rich repeat-containing receptors protein 3; IL-1beta = interleukin-1 beta; PMP = platelet microparticles; FVII = coagulation factor VII.

**Table 1 microorganisms-10-01624-t001:** New frontiers of therapy for *C. difficile* infection-related inflammasome activation and coagulation derangements.

Drug	Class	References
Low molecular weight heparin (LMWH)	To prevent DIC	[54,76]
TLR5-agonist Salmonella-derived flagellin	To boost the microbiome	[73,74]
Thrombomodulin	To prevent DIC	[77]
Recombinant TFPI	To reduce CDI	[78]
Dapansutrile, Hydroxychloroquine, MCC950, CY-09, OLT1177, 3,4,methylenedioxy-β-nitrostyrene (MNS), tranilast	NLRP3 inhibitors	[80,84,85,86]
Oridonin	NLRP3 inhibitors and prevent endothelial dysfunction	[81,86,87]
Pralnacasan	caspase 4 inhibitors	[83]
Belnacasan	caspase 5 inhibitors	[83]
VP20621 and RBX2660	To prevent CDI recurrence	[90,91]
Eculizumab	To prevent endothelial dysfunction	[92]

## Data Availability

Not applicable.

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
