# Peer review of "Clostridium difficile Induced Inflammasome Activation and Coagulation Derangements"

_microorganisms, 2022, doi:10.3390/microorganisms10081624_

Round 1
Reviewer 1 Report
The authors tried to make a Review on "Clostridium Difficile induced inflammasome activation and coagulation derangements". However, I have serious doubts that such a topic can be developed properly in 9 pages (MDPI format=2/3 used from a page). Few of my main suggestions related to this manuscript are mentioned bellow:
In the Title: Clostridium difficile (in italics). All Latin names in the paper must be written according to international rules. Please check and revise
Keywords must reflect the main characteristic words of the paper (usually reflected also by the title) in the best way to increase the paper's relevance and chances to be find when searching it after key words. So, for the actual title, I suggest the following keywords: Clostridium difficile; induced inflammasome activation; coagulation derangements; enterocolitis; sepsis.
Never using abbreviations as Keywords. They are not providing a good visibility when other authors are searching for papers in the topic.
References must be inserted in the main text in brackets, not in parenthesis, according to the Instructions for authors https://www.mdpi.com/journal/microorganisms/instructions , which are given to be followed and applied, not being optionally.
At the final of Introduction section, please add the aim of the study. What makes special this study? Which is its novelty character or its special aspects? Why have the author chosen this topic? What differentiate this paper from others already published in the same/similar topic? Numerous studies are done in this field, so it is important to emphasize what does your study brings special.
The structure for a Review cannot follow the structure recommended for Articles. The titles can be renamed more proper with their content.
- Materials and Methods section can be renamed as 2. Methodology. As the authors have stated that this is a Review, a PRISMA flow chartis recommended, instead of the existing paragraph. I suggest checking Page MJ, McKenzie JE, Bossuyt PM, Boutron I, Hoffmann TC, Mulrow CD, et al. The PRISMA 2020 statement: an updated guideline for reporting systematic reviews. BMJ 2021;372:n71. doi: 10.1136/bmj.n71 regarding PRISMA flow chart. Please take care and detail in the best way the inclusion/exclusion criteria used for the literature selection. Include here also the MeSH terms. Do not forget renumbering the following sections.
Moreover, tenths of papers in the field have been not searched, some of them very recent and informative (e.g. I suggest checking and referring to https://doi.org/10.3390/su12114439 ; https://doi.org/10.3390/healthcare8030352 )
Results section does not present any results, but data from literature. Please rename accordingly.
Tabulated part is missing.
Figures – only one, blurred.
Figures describing mechanisms, pathways, or other important aspects the authors consider must be added.
Section 4.2. I suggest a Table related to therapies. Last column – references (Refs). It would be more relevant.
References must be written in the MDPI style, providing all information requested for each ref.
45 references cannot be considered that describing a topic. The authors must extensively improve the information in this paper (developing more each section/subsection), its structure (renaming some sections and adding more sections) and the References part.
Author Response
The authors tried to make a Review on "Clostridium Difficile induced inflammasome activation and coagulation derangements". However, I have serious doubts that such a topic can be developed properly in 9 pages (MDPI format=2/3 used from a page). Few of my main suggestions related to this manuscript are mentioned bellow:
In the Title: Clostridium difficile (in italics). All Latin names in the paper must be written according to international rules. Please check and revise
RE: Many thanks, we have now checked and rewritten names following international rules
Keywords must reflect the main characteristic words of the paper (usually reflected also by the title) in the best way to increase the paper's relevance and chances to be find when searching it after key words. So, for the actual title, I suggest the following keywords: Clostridium difficile; induced inflammasome activation; coagulation derangements; enterocolitis; sepsis.
RE: Keywords have now been replaced with those suggested
Never using abbreviations as Keywords. They are not providing a good visibility when other authors are searching for papers in the topic.
RE: We have now checked and corrected it,thanks
References must be inserted in the main text in brackets, not in parenthesis, according to the Instructions for authors https://www.mdpi.com/journal/microorganisms/instructions , which are given to be followed and applied, not being optionally.
RE: We have now modified as suggested, thanks
At the final of Introduction section, please add the aim of the study. What makes special this study? Which is its novelty character or its special aspects? Why have the author chosen this topic? What differentiate this paper from others already published in the same/similar topic? Numerous studies are done in this field, so it is important to emphasize what does your study brings special.
Re We have now specified the aims of the study and the novelty of the topic, as suggested
The structure for a Review cannot follow the structure recommended for Articles. The titles can be renamed more proper with their content.
RE: We have now revised it accordingly, thanks
Materials and Methods section can be renamed as 2. Methodology. As the authors have stated that this is a Review, a PRISMA flow chart is recommended, instead of the existing paragraph. I suggest checking Page MJ, McKenzie JE, Bossuyt PM, Boutron I, Hoffmann TC, Mulrow CD, et al. The PRISMA 2020 statement: an updated guideline for reporting systematic reviews. BMJ 2021;372:n71. doi: 10.1136/bmj.n71 regarding PRISMA flow chart. Please take care and detail in the best way the inclusion/exclusion criteria used for the literature selection. Include here also the MeSH terms. Do not forget renumbering the following sections.
RE: Sections have now been renamed. This is a narrative review, we have thus not adopted PRISMA criteria This is reported in the section 2, where we now better described the methodology adopted. Narrative reviews allow the inclusion of large quantities of papers dealing with potentially every possible aspect of a given topic, without necessarily performing a thorough analysis of the statistical evidence of the analyzed data, due to the variety of evidences available on this topic ,focused on different aspects we have chosen this form of review.
Moreover, tenths of papers in the field have been not searched, some of them very recent and informative (e.g. I suggest checking and referring to https://doi.org/10.3390/su12114439 ; https://doi.org/10.3390/healthcare8030352 )
RE: We have now expanded our search and included more relevant papers
Results section does not present any results, but data from literature. Please rename accordingly.
RE: The section has now been renamed.
Tabulated part is missing.
RE: We have now included one table,thanks
Figures – only one, blurred.
RE: We included six figures,thanks
Figures describing mechanisms, pathways, or other important aspects the authors consider must be added.
RE: Figures have been added
Section 4.2. I suggest a Table related to therapies. Last column – references (Refs). It would be more relevant.
Re; We have now added the suggested table
References must be written in the MDPI style, providing all information requested for each ref.
RE: References have now been corrected
45 references cannot be considered that describing a topic. The authors must extensively improve the information in this paper (developing more each section/subsection), its structure (renaming some sections and adding more sections) and the References part.
Re: We have now modified the paper according to suggestions
The authors tried to make a Review on "Clostridium Difficile induced inflammasome activation and coagulation derangements". However, I have serious doubts that such a topic can be developed properly in 9 pages (MDPI format=2/3 used from a page). Few of my main suggestions related to this manuscript are mentioned bellow:
In the Title: Clostridium difficile (in italics). All Latin names in the paper must be written according to international rules. Please check and revise
RE: Many thanks, we have now checked and rewritten names following international rules
Keywords must reflect the main characteristic words of the paper (usually reflected also by the title) in the best way to increase the paper's relevance and chances to be find when searching it after key words. So, for the actual title, I suggest the following keywords: Clostridium difficile; induced inflammasome activation; coagulation derangements; enterocolitis; sepsis.
RE: Keywords have now been replaced with those suggested
Never using abbreviations as Keywords. They are not providing a good visibility when other authors are searching for papers in the topic.
RE: We have now checked and corrected it,thanks
References must be inserted in the main text in brackets, not in parenthesis, according to the Instructions for authors https://www.mdpi.com/journal/microorganisms/instructions , which are given to be followed and applied, not being optionally.
RE: We have now modified as suggested, thanks
At the final of Introduction section, please add the aim of the study. What makes special this study? Which is its novelty character or its special aspects? Why have the author chosen this topic? What differentiate this paper from others already published in the same/similar topic? Numerous studies are done in this field, so it is important to emphasize what does your study brings special.
Re We have now specified the aims of the study and the novelty of the topic, as suggested
The structure for a Review cannot follow the structure recommended for Articles. The titles can be renamed more proper with their content.
RE: We have now revised it accordingly, thanks
Materials and Methods section can be renamed as 2. Methodology. As the authors have stated that this is a Review, a PRISMA flow chart is recommended, instead of the existing paragraph. I suggest checking Page MJ, McKenzie JE, Bossuyt PM, Boutron I, Hoffmann TC, Mulrow CD, et al. The PRISMA 2020 statement: an updated guideline for reporting systematic reviews. BMJ 2021;372:n71. doi: 10.1136/bmj.n71 regarding PRISMA flow chart. Please take care and detail in the best way the inclusion/exclusion criteria used for the literature selection. Include here also the MeSH terms. Do not forget renumbering the following sections.
RE: Sections have now been renamed. This is a narrative review, we have thus not adopted PRISMA criteria This is reported in the section 2, where we now better described the methodology adopted. Narrative reviews allow the inclusion of large quantities of papers dealing with potentially every possible aspect of a given topic, without necessarily performing a thorough analysis of the statistical evidence of the analyzed data, due to the variety of evidences available on this topic ,focused on different aspects we have chosen this form of review.
Moreover, tenths of papers in the field have been not searched, some of them very recent and informative (e.g. I suggest checking and referring to https://doi.org/10.3390/su12114439 ; https://doi.org/10.3390/healthcare8030352 )
RE: We have now expanded our search and included more relevant papers
Results section does not present any results, but data from literature. Please rename accordingly.
RE: The section has now been renamed.
Tabulated part is missing.
RE: We have now included one table,thanks
Figures – only one, blurred.
RE: We included six figures,thanks
Figures describing mechanisms, pathways, or other important aspects the authors consider must be added.
RE: Figures have been added
Section 4.2. I suggest a Table related to therapies. Last column – references (Refs). It would be more relevant.
Re; We have now added the suggested table
References must be written in the MDPI style, providing all information requested for each ref.
RE: References have now been corrected
45 references cannot be considered that describing a topic. The authors must extensively improve the information in this paper (developing more each section/subsection), its structure (renaming some sections and adding more sections) and the References part.
Re: We have now modified the paper according to suggestions
The authors tried to make a Review on "Clostridium Difficile induced inflammasome activation and coagulation derangements". However, I have serious doubts that such a topic can be developed properly in 9 pages (MDPI format=2/3 used from a page). Few of my main suggestions related to this manuscript are mentioned bellow:
In the Title: Clostridium difficile (in italics). All Latin names in the paper must be written according to international rules. Please check and revise
RE: Many thanks, we have now checked and rewritten names following international rules
Keywords must reflect the main characteristic words of the paper (usually reflected also by the title) in the best way to increase the paper's relevance and chances to be find when searching it after key words. So, for the actual title, I suggest the following keywords: Clostridium difficile; induced inflammasome activation; coagulation derangements; enterocolitis; sepsis.
RE: Keywords have now been replaced with those suggested
Never using abbreviations as Keywords. They are not providing a good visibility when other authors are searching for papers in the topic.
RE: We have now checked and corrected it,thanks
References must be inserted in the main text in brackets, not in parenthesis, according to the Instructions for authors https://www.mdpi.com/journal/microorganisms/instructions , which are given to be followed and applied, not being optionally.
RE: We have now modified as suggested, thanks
At the final of Introduction section, please add the aim of the study. What makes special this study? Which is its novelty character or its special aspects? Why have the author chosen this topic? What differentiate this paper from others already published in the same/similar topic? Numerous studies are done in this field, so it is important to emphasize what does your study brings special.
Re We have now specified the aims of the study and the novelty of the topic, as suggested
The structure for a Review cannot follow the structure recommended for Articles. The titles can be renamed more proper with their content.
RE: We have now revised it accordingly, thanks
Materials and Methods section can be renamed as 2. Methodology. As the authors have stated that this is a Review, a PRISMA flow chart is recommended, instead of the existing paragraph. I suggest checking Page MJ, McKenzie JE, Bossuyt PM, Boutron I, Hoffmann TC, Mulrow CD, et al. The PRISMA 2020 statement: an updated guideline for reporting systematic reviews. BMJ 2021;372:n71. doi: 10.1136/bmj.n71 regarding PRISMA flow chart. Please take care and detail in the best way the inclusion/exclusion criteria used for the literature selection. Include here also the MeSH terms. Do not forget renumbering the following sections.
RE: Sections have now been renamed. This is a narrative review, we have thus not adopted PRISMA criteria This is reported in the section 2, where we now better described the methodology adopted. Narrative reviews allow the inclusion of large quantities of papers dealing with potentially every possible aspect of a given topic, without necessarily performing a thorough analysis of the statistical evidence of the analyzed data, due to the variety of evidences available on this topic ,focused on different aspects we have chosen this form of review.
Moreover, tenths of papers in the field have been not searched, some of them very recent and informative (e.g. I suggest checking and referring to https://doi.org/10.3390/su12114439 ; https://doi.org/10.3390/healthcare8030352 )
RE: We have now expanded our search and included more relevant papers
Results section does not present any results, but data from literature. Please rename accordingly.
RE: The section has now been renamed.
Tabulated part is missing.
RE: We have now included one table,thanks
Figures – only one, blurred.
RE: We included six figures,thanks
Figures describing mechanisms, pathways, or other important aspects the authors consider must be added.
RE: Figures have been added
Section 4.2. I suggest a Table related to therapies. Last column – references (Refs). It would be more relevant.
Re; We have now added the suggested table
References must be written in the MDPI style, providing all information requested for each ref.
RE: References have now been corrected
45 references cannot be considered that describing a topic. The authors must extensively improve the information in this paper (developing more each section/subsection), its structure (renaming some sections and adding more sections) and the References part.
Re: We have now modified the paper according to suggestions
The authors tried to make a Review on "Clostridium Difficile induced inflammasome activation and coagulation derangements". However, I have serious doubts that such a topic can be developed properly in 9 pages (MDPI format=2/3 used from a page). Few of my main suggestions related to this manuscript are mentioned bellow:
In the Title: Clostridium difficile (in italics). All Latin names in the paper must be written according to international rules. Please check and revise
RE: Many thanks, we have now checked and rewritten names following international rules
Keywords must reflect the main characteristic words of the paper (usually reflected also by the title) in the best way to increase the paper's relevance and chances to be find when searching it after key words. So, for the actual title, I suggest the following keywords: Clostridium difficile; induced inflammasome activation; coagulation derangements; enterocolitis; sepsis.
RE: Keywords have now been replaced with those suggested
Never using abbreviations as Keywords. They are not providing a good visibility when other authors are searching for papers in the topic.
RE: We have now checked and corrected it,thanks
References must be inserted in the main text in brackets, not in parenthesis, according to the Instructions for authors https://www.mdpi.com/journal/microorganisms/instructions , which are given to be followed and applied, not being optionally.
RE: We have now modified as suggested, thanks
At the final of Introduction section, please add the aim of the study. What makes special this study? Which is its novelty character or its special aspects? Why have the author chosen this topic? What differentiate this paper from others already published in the same/similar topic? Numerous studies are done in this field, so it is important to emphasize what does your study brings special.
Re We have now specified the aims of the study and the novelty of the topic, as suggested
The structure for a Review cannot follow the structure recommended for Articles. The titles can be renamed more proper with their content.
RE: We have now revised it accordingly, thanks
Materials and Methods section can be renamed as 2. Methodology. As the authors have stated that this is a Review, a PRISMA flow chart is recommended, instead of the existing paragraph. I suggest checking Page MJ, McKenzie JE, Bossuyt PM, Boutron I, Hoffmann TC, Mulrow CD, et al. The PRISMA 2020 statement: an updated guideline for reporting systematic reviews. BMJ 2021;372:n71. doi: 10.1136/bmj.n71 regarding PRISMA flow chart. Please take care and detail in the best way the inclusion/exclusion criteria used for the literature selection. Include here also the MeSH terms. Do not forget renumbering the following sections.
RE: Sections have now been renamed. This is a narrative review, we have thus not adopted PRISMA criteria This is reported in the section 2, where we now better described the methodology adopted. Narrative reviews allow the inclusion of large quantities of papers dealing with potentially every possible aspect of a given topic, without necessarily performing a thorough analysis of the statistical evidence of the analyzed data, due to the variety of evidences available on this topic ,focused on different aspects we have chosen this form of review.
Moreover, tenths of papers in the field have been not searched, some of them very recent and informative (e.g. I suggest checking and referring to https://doi.org/10.3390/su12114439 ; https://doi.org/10.3390/healthcare8030352 )
RE: We have now expanded our search and included more relevant papers
Results section does not present any results, but data from literature. Please rename accordingly.
RE: The section has now been renamed.
Tabulated part is missing.
RE: We have now included one table,thanks
Figures – only one, blurred.
RE: We included six figures,thanks
Figures describing mechanisms, pathways, or other important aspects the authors consider must be added.
RE: Figures have been added
Section 4.2. I suggest a Table related to therapies. Last column – references (Refs). It would be more relevant.
Re; We have now added the suggested table
References must be written in the MDPI style, providing all information requested for each ref.
RE: References have now been corrected
45 references cannot be considered that describing a topic. The authors must extensively improve the information in this paper (developing more each section/subsection), its structure (renaming some sections and adding more sections) and the References part.
Re: We have now modified the paper according to suggestions
The authors tried to make a Review on "Clostridium Difficile induced inflammasome activation and coagulation derangements". However, I have serious doubts that such a topic can be developed properly in 9 pages (MDPI format=2/3 used from a page). Few of my main suggestions related to this manuscript are mentioned bellow:
In the Title: Clostridium difficile (in italics). All Latin names in the paper must be written according to international rules. Please check and revise
RE: Many thanks, we have now checked and rewritten names following international rules
Keywords must reflect the main characteristic words of the paper (usually reflected also by the title) in the best way to increase the paper's relevance and chances to be find when searching it after key words. So, for the actual title, I suggest the following keywords: Clostridium difficile; induced inflammasome activation; coagulation derangements; enterocolitis; sepsis.
RE: Keywords have now been replaced with those suggested
Never using abbreviations as Keywords. They are not providing a good visibility when other authors are searching for papers in the topic.
RE: We have now checked and corrected it,thanks
References must be inserted in the main text in brackets, not in parenthesis, according to the Instructions for authors https://www.mdpi.com/journal/microorganisms/instructions , which are given to be followed and applied, not being optionally.
RE: We have now modified as suggested, thanks
At the final of Introduction section, please add the aim of the study. What makes special this study? Which is its novelty character or its special aspects? Why have the author chosen this topic? What differentiate this paper from others already published in the same/similar topic? Numerous studies are done in this field, so it is important to emphasize what does your study brings special.
Re We have now specified the aims of the study and the novelty of the topic, as suggested
The structure for a Review cannot follow the structure recommended for Articles. The titles can be renamed more proper with their content.
RE: We have now revised it accordingly, thanks
Materials and Methods section can be renamed as 2. Methodology. As the authors have stated that this is a Review, a PRISMA flow chart is recommended, instead of the existing paragraph. I suggest checking Page MJ, McKenzie JE, Bossuyt PM, Boutron I, Hoffmann TC, Mulrow CD, et al. The PRISMA 2020 statement: an updated guideline for reporting systematic reviews. BMJ 2021;372:n71. doi: 10.1136/bmj.n71 regarding PRISMA flow chart. Please take care and detail in the best way the inclusion/exclusion criteria used for the literature selection. Include here also the MeSH terms. Do not forget renumbering the following sections.
RE: Sections have now been renamed. This is a narrative review, we have thus not adopted PRISMA criteria This is reported in the section 2, where we now better described the methodology adopted. Narrative reviews allow the inclusion of large quantities of papers dealing with potentially every possible aspect of a given topic, without necessarily performing a thorough analysis of the statistical evidence of the analyzed data, due to the variety of evidences available on this topic ,focused on different aspects we have chosen this form of review.
Moreover, tenths of papers in the field have been not searched, some of them very recent and informative (e.g. I suggest checking and referring to https://doi.org/10.3390/su12114439 ; https://doi.org/10.3390/healthcare8030352 )
RE: We have now expanded our search and included more relevant papers
Results section does not present any results, but data from literature. Please rename accordingly.
RE: The section has now been renamed.
Tabulated part is missing.
RE: We have now included one table,thanks
Figures – only one, blurred.
RE: We included six figures,thanks
Figures describing mechanisms, pathways, or other important aspects the authors consider must be added.
RE: Figures have been added
Section 4.2. I suggest a Table related to therapies. Last column – references (Refs). It would be more relevant.
Re; We have now added the suggested table
References must be written in the MDPI style, providing all information requested for each ref.
RE: References have now been corrected
45 references cannot be considered that describing a topic. The authors must extensively improve the information in this paper (developing more each section/subsection), its structure (renaming some sections and adding more sections) and the References part.
Re: We have now modified the paper according to suggestions
The authors tried to make a Review on "Clostridium Difficile induced inflammasome activation and coagulation derangements". However, I have serious doubts that such a topic can be developed properly in 9 pages (MDPI format=2/3 used from a page). Few of my main suggestions related to this manuscript are mentioned bellow:
In the Title: Clostridium difficile (in italics). All Latin names in the paper must be written according to international rules. Please check and revise
RE: Many thanks, we have now checked and rewritten names following international rules
Keywords must reflect the main characteristic words of the paper (usually reflected also by the title) in the best way to increase the paper's relevance and chances to be find when searching it after key words. So, for the actual title, I suggest the following keywords: Clostridium difficile; induced inflammasome activation; coagulation derangements; enterocolitis; sepsis.
RE: Keywords have now been replaced with those suggested
Never using abbreviations as Keywords. They are not providing a good visibility when other authors are searching for papers in the topic.
RE: We have now checked and corrected it,thanks
References must be inserted in the main text in brackets, not in parenthesis, according to the Instructions for authors https://www.mdpi.com/journal/microorganisms/instructions , which are given to be followed and applied, not being optionally.
RE: We have now modified as suggested, thanks
At the final of Introduction section, please add the aim of the study. What makes special this study? Which is its novelty character or its special aspects? Why have the author chosen this topic? What differentiate this paper from others already published in the same/similar topic? Numerous studies are done in this field, so it is important to emphasize what does your study brings special.
Re We have now specified the aims of the study and the novelty of the topic, as suggested
The structure for a Review cannot follow the structure recommended for Articles. The titles can be renamed more proper with their content.
RE: We have now revised it accordingly, thanks
Materials and Methods section can be renamed as 2. Methodology. As the authors have stated that this is a Review, a PRISMA flow chart is recommended, instead of the existing paragraph. I suggest checking Page MJ, McKenzie JE, Bossuyt PM, Boutron I, Hoffmann TC, Mulrow CD, et al. The PRISMA 2020 statement: an updated guideline for reporting systematic reviews. BMJ 2021;372:n71. doi: 10.1136/bmj.n71 regarding PRISMA flow chart. Please take care and detail in the best way the inclusion/exclusion criteria used for the literature selection. Include here also the MeSH terms. Do not forget renumbering the following sections.
RE: Sections have now been renamed. This is a narrative review, we have thus not adopted PRISMA criteria This is reported in the section 2, where we now better described the methodology adopted. Narrative reviews allow the inclusion of large quantities of papers dealing with potentially every possible aspect of a given topic, without necessarily performing a thorough analysis of the statistical evidence of the analyzed data, due to the variety of evidences available on this topic ,focused on different aspects we have chosen this form of review.
Moreover, tenths of papers in the field have been not searched, some of them very recent and informative (e.g. I suggest checking and referring to https://doi.org/10.3390/su12114439 ; https://doi.org/10.3390/healthcare8030352 )
RE: We have now expanded our search and included more relevant papers
Results section does not present any results, but data from literature. Please rename accordingly.
RE: The section has now been renamed.
Tabulated part is missing.
RE: We have now included one table,thanks
Figures – only one, blurred.
RE: We included six figures,thanks
Figures describing mechanisms, pathways, or other important aspects the authors consider must be added.
RE: Figures have been added
Section 4.2. I suggest a Table related to therapies. Last column – references (Refs). It would be more relevant.
Re; We have now added the suggested table
References must be written in the MDPI style, providing all information requested for each ref.
RE: References have now been corrected
45 references cannot be considered that describing a topic. The authors must extensively improve the information in this paper (developing more each section/subsection), its structure (renaming some sections and adding more sections) and the References part.
Re: We have now modified the paper according to suggestions
Reviewer 2 Report
Mattana and colleagues present an interesting draft focusing on the interplay between inflammasome canonical pathway and hemostasis balance under the presence of Clostridium Difficile-related/induced pathology. This is a novel and well-focused area of medicine, thus, every new contribution to the field is strongly appreciated.
After a carefully conducted review of this paper, I must admit that in its current form it cannot be published, albeit, I strongly encourage authors for extensive revision and resubmission. Please let me explain my major concerns:
The authors discuss a very broad range of interdisciplinary areas - the intersection of hematology, microbiology, and immunology. I do not agree that putting the particular terms in the PubMed database gives so low number of relevant references.
Moving further, there is a very limited amount of information given regarding the following PRISMA guidelines and description of the searching strategy, which is so crucial for review papers. The Author should include at least: Data sources and searches, Study eligibility criteria, Study selection process, Data extraction, and study quality assessment (assessing the risk of bias (ROB) for each included study), Data synthesis. MeSH terms (in addition/replacement of keywords) are necessary to be included. For each step, it is necessary to explain to the reader with pictures or tables. It is necessary to explain what was drawn at each step to lead to the result. Moreover, a figure showing the PRISMA-based workflow must be drawn accordingly to the Prisma schema. After that, a discussion is valuable.
I also strongly encourage the Authors to search additional search engines - what is the golden rule/principal rule of avoiding biased Review paper preparation. I recommend using at least one additional database - preferably Scopus or Google Scholar.
Moving forward, since the title says "inflammasome" there is a very large amount of crucial information related to non-canonical inflammasome pathway - Casp-11-GSDMD axis. It has been shown (doi: 10.1084/jem.20172060 ; https://www.nature.com/articles/s41467-018-07386-5) that this particular axis is crucial for Clostridium d. induced hyperinflammation. This section must be introduced.
I also appreciate the efforts to visualize the discussed phenomena, however, I would like to see newly introduced two figures - first for NLRP3 dependent pathway and its interplay with hemostasis and the second one for inflammasome-dependent platelets activation - moreover, the insight into this interplay between NLRP-3 <=> platelets activation must be expanded and enriched.
I am also missing a very important part of the whole picture - there is no information given about the possible role of inflammasome on the fibrinolysis pathway that is opposite to coagulation, thus, might over-counter discussed effects.
What is more, the Authors are right when discussing the role of neutrophil activation during the inflammatory state - what is the role of NETs - indeed, NETs are a key player when propagating prothrombotic state via inflammasome activation.
Please also include recent clinical trials conducted to prevent Clostridium infection and its poor outcomes.
Please revise the language used in the manuscript. Please use native speakers' help to improve grammatical structure as well as vocabulary throughout the paper. Please do not mix American and British English.
Please remove the Italian language from the Reference section.
To sum up - I am supportive of the publication, albeit, after deeply conducted revise the paper.
Best.
Author Response
Mattana and colleagues present an interesting draft focusing on the interplay between inflammasome canonical pathway and hemostasis balance under the presence of Clostridium Difficile-related/induced pathology. This is a novel and well-focused area of medicine, thus, every new contribution to the field is strongly appreciated.
After a carefully conducted review of this paper, I must admit that in its current form it cannot be published, albeit, I strongly encourage authors for extensive revision and resubmission. Please let me explain my major concerns:
The authors discuss a very broad range of interdisciplinary areas - the intersection of hematology, microbiology, and immunology. I do not agree that putting the particular terms in the PubMed database gives so low number of relevant references.
Ans: We have now included more papers relevant to ther topic,thank you
Moving further, there is a very limited amount of information given regarding the following PRISMA guidelines and description of the searching strategy, which is so crucial for review papers. The Author should include at least: Data sources and searches, Study eligibility criteria, Study selection process, Data extraction, and study quality assessment (assessing the risk of bias (ROB) for each included study), Data synthesis. MeSH terms (in addition/replacement of keywords) are necessary to be included. For each step, it is necessary to explain to the reader with pictures or tables. It is necessary to explain what was drawn at each step to lead to the result. Moreover, a figure showing the PRISMA-based workflow must be drawn accordingly to the Prisma schema. After that, a discussion is valuable.
RE: MeSH terms searched are now reported in Methodology section. We performed a narrative review , thus we did not follow PRISMA criteria, we have now explained in detail in the Methodology section the reasons of this choice
I also strongly encourage the Authors to search additional search engines - what is the golden rule/principal rule of avoiding biased Review paper preparation. I recommend using at least one additional database - preferably Scopus or Google Scholar.
RE: We have now checked and searched also for Google Scholar and Scopus
Moving forward, since the title says "inflammasome" there is a very large amount of crucial information related to non-canonical inflammasome pathway - Casp-11-GSDMD axis. It has been shown (doi: 10.1084/jem.20172060 ; https://www.nature.com/articles/s41467-018-07386-5) that this particular axis is crucial for Clostridium d. induced hyperinflammation. This section must be introduced.
- Thanks, this section has now been added as suggested
I also appreciate the efforts to visualize the discussed phenomena, however, I would like to see newly introduced two figures - first for NLRP3 dependent pathway and its interplay with hemostasis and the second one for inflammasome-dependent platelets activation - moreover, the insight into this interplay between NLRP-3 <=> platelets activation must be expanded and enriched.
RE: Figures have now been added
I am also missing a very important part of the whole picture - there is no information given about the possible role of inflammasome on the fibrinolysis pathway that is opposite to coagulation, thus, might over-counter discussed effects.
RE Thanks, this section has now been added, although there is very little data available for this interaction in the specific setting of CDI which remains the main topic of the paper
What is more, the Authors are right when discussing the role of neutrophil activation during the inflammatory state - what is the role of NETs - indeed, NETs are a key player when propagating prothrombotic state via inflammasome activation.
RE: The section has now been added as suggested
Please also include recent clinical trials conducted to prevent Clostridium infection and its poor outcomes.
RE: Suggested trials have now been added as suggested
Please revise the language used in the manuscript. Please use native speakers' help to improve grammatical structure as well as vocabulary throughout the paper. Please do not mix American and British English.
RE: We have now revised English according to comments
To sum up - I am supportive of the publication, albeit, after deeply conducted revise the paper.
Best.
RE: Many thanks, we hope the revised version of the manuscript will meet your expectations
Mattana and colleagues present an interesting draft focusing on the interplay between inflammasome canonical pathway and hemostasis balance under the presence of Clostridium Difficile-related/induced pathology. This is a novel and well-focused area of medicine, thus, every new contribution to the field is strongly appreciated.
After a carefully conducted review of this paper, I must admit that in its current form it cannot be published, albeit, I strongly encourage authors for extensive revision and resubmission. Please let me explain my major concerns:
The authors discuss a very broad range of interdisciplinary areas - the intersection of hematology, microbiology, and immunology. I do not agree that putting the particular terms in the PubMed database gives so low number of relevant references.
Ans: We have now included more papers relevant to ther topic,thank you
Moving further, there is a very limited amount of information given regarding the following PRISMA guidelines and description of the searching strategy, which is so crucial for review papers. The Author should include at least: Data sources and searches, Study eligibility criteria, Study selection process, Data extraction, and study quality assessment (assessing the risk of bias (ROB) for each included study), Data synthesis. MeSH terms (in addition/replacement of keywords) are necessary to be included. For each step, it is necessary to explain to the reader with pictures or tables. It is necessary to explain what was drawn at each step to lead to the result. Moreover, a figure showing the PRISMA-based workflow must be drawn accordingly to the Prisma schema. After that, a discussion is valuable.
RE: MeSH terms searched are now reported in Methodology section. We performed a narrative review , thus we did not follow PRISMA criteria, we have now explained in detail in the Methodology section the reasons of this choice
I also strongly encourage the Authors to search additional search engines - what is the golden rule/principal rule of avoiding biased Review paper preparation. I recommend using at least one additional database - preferably Scopus or Google Scholar.
RE: We have now checked and searched also for Google Scholar and Scopus
Moving forward, since the title says "inflammasome" there is a very large amount of crucial information related to non-canonical inflammasome pathway - Casp-11-GSDMD axis. It has been shown (doi: 10.1084/jem.20172060 ; https://www.nature.com/articles/s41467-018-07386-5) that this particular axis is crucial for Clostridium d. induced hyperinflammation. This section must be introduced.
- Thanks, this section has now been added as suggested
I also appreciate the efforts to visualize the discussed phenomena, however, I would like to see newly introduced two figures - first for NLRP3 dependent pathway and its interplay with hemostasis and the second one for inflammasome-dependent platelets activation - moreover, the insight into this interplay between NLRP-3 <=> platelets activation must be expanded and enriched.
RE: Figures have now been added
I am also missing a very important part of the whole picture - there is no information given about the possible role of inflammasome on the fibrinolysis pathway that is opposite to coagulation, thus, might over-counter discussed effects.
RE Thanks, this section has now been added, although there is very little data available for this interaction in the specific setting of CDI which remains the main topic of the paper
What is more, the Authors are right when discussing the role of neutrophil activation during the inflammatory state - what is the role of NETs - indeed, NETs are a key player when propagating prothrombotic state via inflammasome activation.
RE: The section has now been added as suggested
Please also include recent clinical trials conducted to prevent Clostridium infection and its poor outcomes.
RE: Suggested trials have now been added as suggested
Please revise the language used in the manuscript. Please use native speakers' help to improve grammatical structure as well as vocabulary throughout the paper. Please do not mix American and British English.
RE: We have now revised English according to comments
To sum up - I am supportive of the publication, albeit, after deeply conducted revise the paper.
Best.
RE: Many thanks, we hope the revised version of the manuscript will meet your expectations
Mattana and colleagues present an interesting draft focusing on the interplay between inflammasome canonical pathway and hemostasis balance under the presence of Clostridium Difficile-related/induced pathology. This is a novel and well-focused area of medicine, thus, every new contribution to the field is strongly appreciated.
After a carefully conducted review of this paper, I must admit that in its current form it cannot be published, albeit, I strongly encourage authors for extensive revision and resubmission. Please let me explain my major concerns:
The authors discuss a very broad range of interdisciplinary areas - the intersection of hematology, microbiology, and immunology. I do not agree that putting the particular terms in the PubMed database gives so low number of relevant references.
Ans: We have now included more papers relevant to ther topic,thank you
Moving further, there is a very limited amount of information given regarding the following PRISMA guidelines and description of the searching strategy, which is so crucial for review papers. The Author should include at least: Data sources and searches, Study eligibility criteria, Study selection process, Data extraction, and study quality assessment (assessing the risk of bias (ROB) for each included study), Data synthesis. MeSH terms (in addition/replacement of keywords) are necessary to be included. For each step, it is necessary to explain to the reader with pictures or tables. It is necessary to explain what was drawn at each step to lead to the result. Moreover, a figure showing the PRISMA-based workflow must be drawn accordingly to the Prisma schema. After that, a discussion is valuable.
RE: MeSH terms searched are now reported in Methodology section. We performed a narrative review , thus we did not follow PRISMA criteria, we have now explained in detail in the Methodology section the reasons of this choice
I also strongly encourage the Authors to search additional search engines - what is the golden rule/principal rule of avoiding biased Review paper preparation. I recommend using at least one additional database - preferably Scopus or Google Scholar.
RE: We have now checked and searched also for Google Scholar and Scopus
Moving forward, since the title says "inflammasome" there is a very large amount of crucial information related to non-canonical inflammasome pathway - Casp-11-GSDMD axis. It has been shown (doi: 10.1084/jem.20172060 ; https://www.nature.com/articles/s41467-018-07386-5) that this particular axis is crucial for Clostridium d. induced hyperinflammation. This section must be introduced.
- Thanks, this section has now been added as suggested
I also appreciate the efforts to visualize the discussed phenomena, however, I would like to see newly introduced two figures - first for NLRP3 dependent pathway and its interplay with hemostasis and the second one for inflammasome-dependent platelets activation - moreover, the insight into this interplay between NLRP-3 <=> platelets activation must be expanded and enriched.
RE: Figures have now been added
I am also missing a very important part of the whole picture - there is no information given about the possible role of inflammasome on the fibrinolysis pathway that is opposite to coagulation, thus, might over-counter discussed effects.
RE Thanks, this section has now been added, although there is very little data available for this interaction in the specific setting of CDI which remains the main topic of the paper
What is more, the Authors are right when discussing the role of neutrophil activation during the inflammatory state - what is the role of NETs - indeed, NETs are a key player when propagating prothrombotic state via inflammasome activation.
RE: The section has now been added as suggested
Please also include recent clinical trials conducted to prevent Clostridium infection and its poor outcomes.
RE: Suggested trials have now been added as suggested
Please revise the language used in the manuscript. Please use native speakers' help to improve grammatical structure as well as vocabulary throughout the paper. Please do not mix American and British English.
RE: We have now revised English according to comments
To sum up - I am supportive of the publication, albeit, after deeply conducted revise the paper.
Best.
RE: Many thanks, we hope the revised version of the manuscript will meet your expectations
Mattana and colleagues present an interesting draft focusing on the interplay between inflammasome canonical pathway and hemostasis balance under the presence of Clostridium Difficile-related/induced pathology. This is a novel and well-focused area of medicine, thus, every new contribution to the field is strongly appreciated.
After a carefully conducted review of this paper, I must admit that in its current form it cannot be published, albeit, I strongly encourage authors for extensive revision and resubmission. Please let me explain my major concerns:
The authors discuss a very broad range of interdisciplinary areas - the intersection of hematology, microbiology, and immunology. I do not agree that putting the particular terms in the PubMed database gives so low number of relevant references.
Ans: We have now included more papers relevant to ther topic,thank you
Moving further, there is a very limited amount of information given regarding the following PRISMA guidelines and description of the searching strategy, which is so crucial for review papers. The Author should include at least: Data sources and searches, Study eligibility criteria, Study selection process, Data extraction, and study quality assessment (assessing the risk of bias (ROB) for each included study), Data synthesis. MeSH terms (in addition/replacement of keywords) are necessary to be included. For each step, it is necessary to explain to the reader with pictures or tables. It is necessary to explain what was drawn at each step to lead to the result. Moreover, a figure showing the PRISMA-based workflow must be drawn accordingly to the Prisma schema. After that, a discussion is valuable.
RE: MeSH terms searched are now reported in Methodology section. We performed a narrative review , thus we did not follow PRISMA criteria, we have now explained in detail in the Methodology section the reasons of this choice
I also strongly encourage the Authors to search additional search engines - what is the golden rule/principal rule of avoiding biased Review paper preparation. I recommend using at least one additional database - preferably Scopus or Google Scholar.
RE: We have now checked and searched also for Google Scholar and Scopus
Moving forward, since the title says "inflammasome" there is a very large amount of crucial information related to non-canonical inflammasome pathway - Casp-11-GSDMD axis. It has been shown (doi: 10.1084/jem.20172060 ; https://www.nature.com/articles/s41467-018-07386-5) that this particular axis is crucial for Clostridium d. induced hyperinflammation. This section must be introduced.
- Thanks, this section has now been added as suggested
I also appreciate the efforts to visualize the discussed phenomena, however, I would like to see newly introduced two figures - first for NLRP3 dependent pathway and its interplay with hemostasis and the second one for inflammasome-dependent platelets activation - moreover, the insight into this interplay between NLRP-3 <=> platelets activation must be expanded and enriched.
RE: Figures have now been added
I am also missing a very important part of the whole picture - there is no information given about the possible role of inflammasome on the fibrinolysis pathway that is opposite to coagulation, thus, might over-counter discussed effects.
RE Thanks, this section has now been added, although there is very little data available for this interaction in the specific setting of CDI which remains the main topic of the paper
What is more, the Authors are right when discussing the role of neutrophil activation during the inflammatory state - what is the role of NETs - indeed, NETs are a key player when propagating prothrombotic state via inflammasome activation.
RE: The section has now been added as suggested
Please also include recent clinical trials conducted to prevent Clostridium infection and its poor outcomes.
RE: Suggested trials have now been added as suggested
Please revise the language used in the manuscript. Please use native speakers' help to improve grammatical structure as well as vocabulary throughout the paper. Please do not mix American and British English.
RE: We have now revised English according to comments
To sum up - I am supportive of the publication, albeit, after deeply conducted revise the paper.
Best.
RE: Many thanks, we hope the revised version of the manuscript will meet your expectations
Mattana and colleagues present an interesting draft focusing on the interplay between inflammasome canonical pathway and hemostasis balance under the presence of Clostridium Difficile-related/induced pathology. This is a novel and well-focused area of medicine, thus, every new contribution to the field is strongly appreciated.
After a carefully conducted review of this paper, I must admit that in its current form it cannot be published, albeit, I strongly encourage authors for extensive revision and resubmission. Please let me explain my major concerns:
The authors discuss a very broad range of interdisciplinary areas - the intersection of hematology, microbiology, and immunology. I do not agree that putting the particular terms in the PubMed database gives so low number of relevant references.
Ans: We have now included more papers relevant to ther topic,thank you
Moving further, there is a very limited amount of information given regarding the following PRISMA guidelines and description of the searching strategy, which is so crucial for review papers. The Author should include at least: Data sources and searches, Study eligibility criteria, Study selection process, Data extraction, and study quality assessment (assessing the risk of bias (ROB) for each included study), Data synthesis. MeSH terms (in addition/replacement of keywords) are necessary to be included. For each step, it is necessary to explain to the reader with pictures or tables. It is necessary to explain what was drawn at each step to lead to the result. Moreover, a figure showing the PRISMA-based workflow must be drawn accordingly to the Prisma schema. After that, a discussion is valuable.
RE: MeSH terms searched are now reported in Methodology section. We performed a narrative review , thus we did not follow PRISMA criteria, we have now explained in detail in the Methodology section the reasons of this choice
I also strongly encourage the Authors to search additional search engines - what is the golden rule/principal rule of avoiding biased Review paper preparation. I recommend using at least one additional database - preferably Scopus or Google Scholar.
RE: We have now checked and searched also for Google Scholar and Scopus
Moving forward, since the title says "inflammasome" there is a very large amount of crucial information related to non-canonical inflammasome pathway - Casp-11-GSDMD axis. It has been shown (doi: 10.1084/jem.20172060 ; https://www.nature.com/articles/s41467-018-07386-5) that this particular axis is crucial for Clostridium d. induced hyperinflammation. This section must be introduced.
- Thanks, this section has now been added as suggested
I also appreciate the efforts to visualize the discussed phenomena, however, I would like to see newly introduced two figures - first for NLRP3 dependent pathway and its interplay with hemostasis and the second one for inflammasome-dependent platelets activation - moreover, the insight into this interplay between NLRP-3 <=> platelets activation must be expanded and enriched.
RE: Figures have now been added
I am also missing a very important part of the whole picture - there is no information given about the possible role of inflammasome on the fibrinolysis pathway that is opposite to coagulation, thus, might over-counter discussed effects.
RE Thanks, this section has now been added, although there is very little data available for this interaction in the specific setting of CDI which remains the main topic of the paper
What is more, the Authors are right when discussing the role of neutrophil activation during the inflammatory state - what is the role of NETs - indeed, NETs are a key player when propagating prothrombotic state via inflammasome activation.
RE: The section has now been added as suggested
Please also include recent clinical trials conducted to prevent Clostridium infection and its poor outcomes.
RE: Suggested trials have now been added as suggested
Please revise the language used in the manuscript. Please use native speakers' help to improve grammatical structure as well as vocabulary throughout the paper. Please do not mix American and British English.
RE: We have now revised English according to comments
To sum up - I am supportive of the publication, albeit, after deeply conducted revise the paper.
Best.
RE: Many thanks, we hope the revised version of the manuscript will meet your expectations
Mattana and colleagues present an interesting draft focusing on the interplay between inflammasome canonical pathway and hemostasis balance under the presence of Clostridium Difficile-related/induced pathology. This is a novel and well-focused area of medicine, thus, every new contribution to the field is strongly appreciated.
After a carefully conducted review of this paper, I must admit that in its current form it cannot be published, albeit, I strongly encourage authors for extensive revision and resubmission. Please let me explain my major concerns:
The authors discuss a very broad range of interdisciplinary areas - the intersection of hematology, microbiology, and immunology. I do not agree that putting the particular terms in the PubMed database gives so low number of relevant references.
Ans: We have now included more papers relevant to ther topic,thank you
Moving further, there is a very limited amount of information given regarding the following PRISMA guidelines and description of the searching strategy, which is so crucial for review papers. The Author should include at least: Data sources and searches, Study eligibility criteria, Study selection process, Data extraction, and study quality assessment (assessing the risk of bias (ROB) for each included study), Data synthesis. MeSH terms (in addition/replacement of keywords) are necessary to be included. For each step, it is necessary to explain to the reader with pictures or tables. It is necessary to explain what was drawn at each step to lead to the result. Moreover, a figure showing the PRISMA-based workflow must be drawn accordingly to the Prisma schema. After that, a discussion is valuable.
RE: MeSH terms searched are now reported in Methodology section. We performed a narrative review , thus we did not follow PRISMA criteria, we have now explained in detail in the Methodology section the reasons of this choice
I also strongly encourage the Authors to search additional search engines - what is the golden rule/principal rule of avoiding biased Review paper preparation. I recommend using at least one additional database - preferably Scopus or Google Scholar.
RE: We have now checked and searched also for Google Scholar and Scopus
Moving forward, since the title says "inflammasome" there is a very large amount of crucial information related to non-canonical inflammasome pathway - Casp-11-GSDMD axis. It has been shown (doi: 10.1084/jem.20172060 ; https://www.nature.com/articles/s41467-018-07386-5) that this particular axis is crucial for Clostridium d. induced hyperinflammation. This section must be introduced.
- Thanks, this section has now been added as suggested
I also appreciate the efforts to visualize the discussed phenomena, however, I would like to see newly introduced two figures - first for NLRP3 dependent pathway and its interplay with hemostasis and the second one for inflammasome-dependent platelets activation - moreover, the insight into this interplay between NLRP-3 <=> platelets activation must be expanded and enriched.
RE: Figures have now been added
I am also missing a very important part of the whole picture - there is no information given about the possible role of inflammasome on the fibrinolysis pathway that is opposite to coagulation, thus, might over-counter discussed effects.
RE Thanks, this section has now been added, although there is very little data available for this interaction in the specific setting of CDI which remains the main topic of the paper
What is more, the Authors are right when discussing the role of neutrophil activation during the inflammatory state - what is the role of NETs - indeed, NETs are a key player when propagating prothrombotic state via inflammasome activation.
RE: The section has now been added as suggested
Please also include recent clinical trials conducted to prevent Clostridium infection and its poor outcomes.
RE: Suggested trials have now been added as suggested
Please revise the language used in the manuscript. Please use native speakers' help to improve grammatical structure as well as vocabulary throughout the paper. Please do not mix American and British English.
RE: We have now revised English according to comments
To sum up - I am supportive of the publication, albeit, after deeply conducted revise the paper.
Best.
RE: Many thanks, we hope the revised version of the manuscript will meet your expectations
Round 2
Reviewer 1 Report
Responses are given in a very strange way, being pasted few times.
In the title, Costridium difficile remained not in Italics. It is a Latin names, it must be italicised. Same in the main manuscript, for C. difficile. The entire manuscript must be revised in this regard.
Even I suggested the proper reference for making a PRISMA flow chart for the selected literature, the authors have chosen only to describe the methodology.
No novelty of the text.
Figures were captured directly from word document, where some words were not recognised and remained underlined with red. Figures must be saved in pdf and only after cropping them.
Too much empty space in the manuscript. The aspect is very poor.
Many figures are blurred. They must be cropped from the original ppt and pasted in the manuscript, not saving them in any format that loose clarity.
Author Response
Revisore 1
Responses are given in a very strange way, being pasted few times.
RE: We have mainly accepted reviwer suggestions and modified the text accordingly
In the title, Costridium difficile remained not in Italics. It is a Latin names, it must be italicised. Same in the main manuscript, for C. difficile. The entire manuscript must be revised in this regard.
RE: We have now revised the manuscript with reference to the renaimed “Clostridium difficle”
Even I suggested the proper reference for making a PRISMA flow chart for the selected literature, the authors have chosen only to describe the methodology.
RE: We have not performed a systematic review, but a narrative review, we have explained in the revised version of the manuscript in the materials section the reasons for this choice
No novelty of the text.
RE: We have updated references and added the suggested revisions in the previous revision process, we have done our best to make the text more interesting
Figures were captured directly from word document, where some words were not recognised and remained underlined with red. Figures must be saved in pdf and only after cropping them.
RE: We have now modified and saved figures as suggested
Too much empty space in the manuscript. The aspect is very poor.
RE: We have now reduced the empty space between paragraphs, They were left empty to allow an easier reading of the revised manuscript
Many figures are blurred. They must be cropped from the original ppt and pasted in the manuscript, not saving them in any format that loose clarity.
RE: We have now modified figures accordingly and tried to improve their definition
Revisore 1
Responses are given in a very strange way, being pasted few times.
RE: We have mainly accepted reviwer suggestions and modified the text accordingly
In the title, Costridium difficile remained not in Italics. It is a Latin names, it must be italicised. Same in the main manuscript, for C. difficile. The entire manuscript must be revised in this regard.
RE: We have now revised the manuscript with reference to the renaimed “Clostridium difficle”
Even I suggested the proper reference for making a PRISMA flow chart for the selected literature, the authors have chosen only to describe the methodology.
RE: We have not performed a systematic review, but a narrative review, we have explained in the revised version of the manuscript in the materials section the reasons for this choice
No novelty of the text.
RE: We have updated references and added the suggested revisions in the previous revision process, we have done our best to make the text more interesting
Figures were captured directly from word document, where some words were not recognised and remained underlined with red. Figures must be saved in pdf and only after cropping them.
RE: We have now modified and saved figures as suggested
Too much empty space in the manuscript. The aspect is very poor.
RE: We have now reduced the empty space between paragraphs, They were left empty to allow an easier reading of the revised manuscript
Many figures are blurred. They must be cropped from the original ppt and pasted in the manuscript, not saving them in any format that loose clarity.
RE: We have now modified figures accordingly and tried to improve their definition
Revisore 1
Responses are given in a very strange way, being pasted few times.
RE: We have mainly accepted reviwer suggestions and modified the text accordingly
In the title, Costridium difficile remained not in Italics. It is a Latin names, it must be italicised. Same in the main manuscript, for C. difficile. The entire manuscript must be revised in this regard.
RE: We have now revised the manuscript with reference to the renaimed “Clostridium difficle”
Even I suggested the proper reference for making a PRISMA flow chart for the selected literature, the authors have chosen only to describe the methodology.
RE: We have not performed a systematic review, but a narrative review, we have explained in the revised version of the manuscript in the materials section the reasons for this choice
No novelty of the text.
RE: We have updated references and added the suggested revisions in the previous revision process, we have done our best to make the text more interesting
Figures were captured directly from word document, where some words were not recognised and remained underlined with red. Figures must be saved in pdf and only after cropping them.
RE: We have now modified and saved figures as suggested
Too much empty space in the manuscript. The aspect is very poor.
RE: We have now reduced the empty space between paragraphs, They were left empty to allow an easier reading of the revised manuscript
Many figures are blurred. They must be cropped from the original ppt and pasted in the manuscript, not saving them in any format that loose clarity.
RE: We have now modified figures accordingly and tried to improve their definition
Revisore 1
Responses are given in a very strange way, being pasted few times.
RE: We have mainly accepted reviwer suggestions and modified the text accordingly
In the title, Costridium difficile remained not in Italics. It is a Latin names, it must be italicised. Same in the main manuscript, for C. difficile. The entire manuscript must be revised in this regard.
RE: We have now revised the manuscript with reference to the renaimed “Clostridium difficle”
Even I suggested the proper reference for making a PRISMA flow chart for the selected literature, the authors have chosen only to describe the methodology.
RE: We have not performed a systematic review, but a narrative review, we have explained in the revised version of the manuscript in the materials section the reasons for this choice
No novelty of the text.
RE: We have updated references and added the suggested revisions in the previous revision process, we have done our best to make the text more interesting
Figures were captured directly from word document, where some words were not recognised and remained underlined with red. Figures must be saved in pdf and only after cropping them.
RE: We have now modified and saved figures as suggested
Too much empty space in the manuscript. The aspect is very poor.
RE: We have now reduced the empty space between paragraphs, They were left empty to allow an easier reading of the revised manuscript
Many figures are blurred. They must be cropped from the original ppt and pasted in the manuscript, not saving them in any format that loose clarity.
RE: We have now modified figures accordingly and tried to improve their definition
Revisore 1
Responses are given in a very strange way, being pasted few times.
RE: We have mainly accepted reviwer suggestions and modified the text accordingly
In the title, Costridium difficile remained not in Italics. It is a Latin names, it must be italicised. Same in the main manuscript, for C. difficile. The entire manuscript must be revised in this regard.
RE: We have now revised the manuscript with reference to the renaimed “Clostridium difficle”
Even I suggested the proper reference for making a PRISMA flow chart for the selected literature, the authors have chosen only to describe the methodology.
RE: We have not performed a systematic review, but a narrative review, we have explained in the revised version of the manuscript in the materials section the reasons for this choice
No novelty of the text.
RE: We have updated references and added the suggested revisions in the previous revision process, we have done our best to make the text more interesting
Figures were captured directly from word document, where some words were not recognised and remained underlined with red. Figures must be saved in pdf and only after cropping them.
RE: We have now modified and saved figures as suggested
Too much empty space in the manuscript. The aspect is very poor.
RE: We have now reduced the empty space between paragraphs, They were left empty to allow an easier reading of the revised manuscript
Many figures are blurred. They must be cropped from the original ppt and pasted in the manuscript, not saving them in any format that loose clarity.
RE: We have now modified figures accordingly and tried to improve their definition
Revisore 1
Responses are given in a very strange way, being pasted few times.
RE: We have mainly accepted reviwer suggestions and modified the text accordingly
In the title, Costridium difficile remained not in Italics. It is a Latin names, it must be italicised. Same in the main manuscript, for C. difficile. The entire manuscript must be revised in this regard.
RE: We have now revised the manuscript with reference to the renaimed “Clostridium difficle”
Even I suggested the proper reference for making a PRISMA flow chart for the selected literature, the authors have chosen only to describe the methodology.
RE: We have not performed a systematic review, but a narrative review, we have explained in the revised version of the manuscript in the materials section the reasons for this choice
No novelty of the text.
RE: We have updated references and added the suggested revisions in the previous revision process, we have done our best to make the text more interesting
Figures were captured directly from word document, where some words were not recognised and remained underlined with red. Figures must be saved in pdf and only after cropping them.
RE: We have now modified and saved figures as suggested
Too much empty space in the manuscript. The aspect is very poor.
RE: We have now reduced the empty space between paragraphs, They were left empty to allow an easier reading of the revised manuscript
Many figures are blurred. They must be cropped from the original ppt and pasted in the manuscript, not saving them in any format that loose clarity.
RE: We have now modified figures accordingly and tried to improve their definition
Revisore 1
Responses are given in a very strange way, being pasted few times.
RE: We have mainly accepted reviwer suggestions and modified the text accordingly
In the title, Costridium difficile remained not in Italics. It is a Latin names, it must be italicised. Same in the main manuscript, for C. difficile. The entire manuscript must be revised in this regard.
RE: We have now revised the manuscript with reference to the renaimed “Clostridium difficle”
Even I suggested the proper reference for making a PRISMA flow chart for the selected literature, the authors have chosen only to describe the methodology.
RE: We have not performed a systematic review, but a narrative review, we have explained in the revised version of the manuscript in the materials section the reasons for this choice
No novelty of the text.
RE: We have updated references and added the suggested revisions in the previous revision process, we have done our best to make the text more interesting
Figures were captured directly from word document, where some words were not recognised and remained underlined with red. Figures must be saved in pdf and only after cropping them.
RE: We have now modified and saved figures as suggested
Too much empty space in the manuscript. The aspect is very poor.
RE: We have now reduced the empty space between paragraphs, They were left empty to allow an easier reading of the revised manuscript
Many figures are blurred. They must be cropped from the original ppt and pasted in the manuscript, not saving them in any format that loose clarity.
RE: We have now modified figures accordingly and tried to improve their definition
Revisore 1
Responses are given in a very strange way, being pasted few times.
RE: We have mainly accepted reviwer suggestions and modified the text accordingly
In the title, Costridium difficile remained not in Italics. It is a Latin names, it must be italicised. Same in the main manuscript, for C. difficile. The entire manuscript must be revised in this regard.
RE: We have now revised the manuscript with reference to the renaimed “Clostridium difficle”
Even I suggested the proper reference for making a PRISMA flow chart for the selected literature, the authors have chosen only to describe the methodology.
RE: We have not performed a systematic review, but a narrative review, we have explained in the revised version of the manuscript in the materials section the reasons for this choice
No novelty of the text.
RE: We have updated references and added the suggested revisions in the previous revision process, we have done our best to make the text more interesting
Figures were captured directly from word document, where some words were not recognised and remained underlined with red. Figures must be saved in pdf and only after cropping them.
RE: We have now modified and saved figures as suggested
Too much empty space in the manuscript. The aspect is very poor.
RE: We have now reduced the empty space between paragraphs, They were left empty to allow an easier reading of the revised manuscript
Many figures are blurred. They must be cropped from the original ppt and pasted in the manuscript, not saving them in any format that loose clarity.
RE: We have now modified figures accordingly and tried to improve their definition
Revisore 1
Responses are given in a very strange way, being pasted few times.
RE: We have mainly accepted reviwer suggestions and modified the text accordingly
In the title, Costridium difficile remained not in Italics. It is a Latin names, it must be italicised. Same in the main manuscript, for C. difficile. The entire manuscript must be revised in this regard.
RE: We have now revised the manuscript with reference to the renaimed “Clostridium difficle”
Even I suggested the proper reference for making a PRISMA flow chart for the selected literature, the authors have chosen only to describe the methodology.
RE: We have not performed a systematic review, but a narrative review, we have explained in the revised version of the manuscript in the materials section the reasons for this choice
No novelty of the text.
RE: We have updated references and added the suggested revisions in the previous revision process, we have done our best to make the text more interesting
Figures were captured directly from word document, where some words were not recognised and remained underlined with red. Figures must be saved in pdf and only after cropping them.
RE: We have now modified and saved figures as suggested
Too much empty space in the manuscript. The aspect is very poor.
RE: We have now reduced the empty space between paragraphs, They were left empty to allow an easier reading of the revised manuscript
Many figures are blurred. They must be cropped from the original ppt and pasted in the manuscript, not saving them in any format that loose clarity.
RE: We have now modified figures accordingly and tried to improve their definition
Revisore 1
Responses are given in a very strange way, being pasted few times.
RE: We have mainly accepted reviwer suggestions and modified the text accordingly
In the title, Costridium difficile remained not in Italics. It is a Latin names, it must be italicised. Same in the main manuscript, for C. difficile. The entire manuscript must be revised in this regard.
RE: We have now revised the manuscript with reference to the renaimed “Clostridium difficle”
Even I suggested the proper reference for making a PRISMA flow chart for the selected literature, the authors have chosen only to describe the methodology.
RE: We have not performed a systematic review, but a narrative review, we have explained in the revised version of the manuscript in the materials section the reasons for this choice
No novelty of the text.
RE: We have updated references and added the suggested revisions in the previous revision process, we have done our best to make the text more interesting
Figures were captured directly from word document, where some words were not recognised and remained underlined with red. Figures must be saved in pdf and only after cropping them.
RE: We have now modified and saved figures as suggested
Too much empty space in the manuscript. The aspect is very poor.
RE: We have now reduced the empty space between paragraphs, They were left empty to allow an easier reading of the revised manuscript
Many figures are blurred. They must be cropped from the original ppt and pasted in the manuscript, not saving them in any format that loose clarity.
RE: We have now modified figures accordingly and tried to improve their definition
Reviewer 2 Report
The Authors conducted an in-depth revision of the presented paper.
I am really satisfied with the work that has been done as well as appreciate Author's efforts in improving the manuscript.
I think that the last minor which is remaining is to update the paper with two decent works from top journals that have been published during the peer-review process, please refer to:
DOI: 10.1182/blood.2021014552
DOI: 10.1186/s12964-022-00907-2
Once again, congratulations!
Author Response
The Authors conducted an in-depth revision of the presented paper.
I am really satisfied with the work that has been done as well as appreciate Author's efforts in improving the manuscript.
I think that the last minor which is remaining is to update the paper with two decent works from top journals that have been published during the peer-review process, please refer to:
DOI: 10.1182/blood.2021014552
DOI: 10.1186/s12964-022-00907-2
Once again, congratulations!
Re: We thank reviwer for the positive evaluation of the revised manuscript, we have now added the suggested references
The Authors conducted an in-depth revision of the presented paper.
I am really satisfied with the work that has been done as well as appreciate Author's efforts in improving the manuscript.
I think that the last minor which is remaining is to update the paper with two decent works from top journals that have been published during the peer-review process, please refer to:
DOI: 10.1182/blood.2021014552
DOI: 10.1186/s12964-022-00907-2
Once again, congratulations!
Re: We thank reviwer for the positive evaluation of the revised manuscript, we have now added the suggested references
The Authors conducted an in-depth revision of the presented paper.
I am really satisfied with the work that has been done as well as appreciate Author's efforts in improving the manuscript.
I think that the last minor which is remaining is to update the paper with two decent works from top journals that have been published during the peer-review process, please refer to:
DOI: 10.1182/blood.2021014552
DOI: 10.1186/s12964-022-00907-2
Once again, congratulations!
Re: We thank reviwer for the positive evaluation of the revised manuscript, we have now added the suggested references
The Authors conducted an in-depth revision of the presented paper.
I am really satisfied with the work that has been done as well as appreciate Author's efforts in improving the manuscript.
I think that the last minor which is remaining is to update the paper with two decent works from top journals that have been published during the peer-review process, please refer to:
DOI: 10.1182/blood.2021014552
DOI: 10.1186/s12964-022-00907-2
Once again, congratulations!
Re: We thank reviwer for the positive evaluation of the revised manuscript, we have now added the suggested references
The Authors conducted an in-depth revision of the presented paper.
I am really satisfied with the work that has been done as well as appreciate Author's efforts in improving the manuscript.
I think that the last minor which is remaining is to update the paper with two decent works from top journals that have been published during the peer-review process, please refer to:
DOI: 10.1182/blood.2021014552
DOI: 10.1186/s12964-022-00907-2
Once again, congratulations!
Re: We thank reviwer for the positive evaluation of the revised manuscript, we have now added the suggested references
The Authors conducted an in-depth revision of the presented paper.
I am really satisfied with the work that has been done as well as appreciate Author's efforts in improving the manuscript.
I think that the last minor which is remaining is to update the paper with two decent works from top journals that have been published during the peer-review process, please refer to:
DOI: 10.1182/blood.2021014552
DOI: 10.1186/s12964-022-00907-2
Once again, congratulations!
Re: We thank reviewer for the positive evaluation of the revised manuscript, we have now added the suggested references